

# The Climate Benefit of Carbon Sequestration

Carlos A. Sierra[1], Susan E. Crow[2], Martin Heimann[1,3], Holger Metzler[1], and Ernst-Detlef Schulze[1]

[1]Max Planck Institute for Biogeochemistry, 07745 Jena, Germany
[2]University of Hawaii Manoa, Honolulu, HI 96822, USA
[3]University of Helsinki, 00560 Helsinki, Finland

**Correspondence:** Carlos A. Sierra (csierra@bgc-jena.mpg.de)

**Abstract.** Ecosystems play a fundamental role in climate change mitigation by taking up carbon from the atmosphere and storing it for a period of time in organic matter. Although climate impacts of carbon emissions can be quantified by global warming potentials, it is not necessarily clear what are appropriate formal metrics to assess climate benefits of carbon removals by sinks. We introduce here the Climate Benefit of Sequestration (CBS), a metric that quantifies the radiative effect of taking

up carbon dioxide from the atmosphere and retaining it for a period of time in an ecosystem before releasing it back to the atmosphere. To quantify CBS, we also propose a formal definition of carbon sequestration (CS) as the integral of an amount of carbon taken up from the atmosphere stored over the time horizon it remains in an ecosystem. Both metrics incorporate the separate effects of i) inputs (amount of atmospheric carbon removal), and ii) transit time (time of carbon retention) in carbon sinks, which can vary largely for different ecosystems or management types. In three separate examples, we show how to

compute and apply these metrics to compare different carbon management practices in forestry and soils. We believe these metrics can be useful in resolving current controversies about the management of ecosystems for climate change mitigation.

## 1 Introduction

Terrestrial ecosystems exchange carbon with the atmosphere at globally significant quantities, thereby influencing Earth's climate and potentially mitigating warming caused by increasing concentrations of $CO_2$ in the atmosphere. Carbon taken

up during the process of photosynthesis remains stored in the terrestrial biosphere over a range of timescales, from days to millennia; timescales comparable with those of greenhouse gases in the atmosphere (Archer et al., 2009; IPCC, 2014; Joos et al., 2013). During the time carbon is stored in the terrestrial biosphere, it is removed from the radiative forcing effect that occurs in the atmosphere; thus, it is of scientific and policy relevance to understand the timescale of carbon storage in ecosystems; i.e. for how long newly fixed carbon is retained in an ecosystem before it is released back to the atmosphere.

Timescales of element cycling and storage are unambiguously characterized by the concepts of *system age* and *transit time* (Bolin and Rodhe, 1973; Rodhe, 2000; Rasmussen et al., 2016; Sierra et al., 2017; Lu et al., 2018). In a system of multiple interconnected compartments, system age characterizes the time that the mass of an element observed in the system has remained there since its entry. Transit time characterizes the time that it takes element masses to traverse the entire system, from the time of entry until they are released back to the external environment (Sierra et al., 2017). Both metrics are excellent





system-level diagnostics of the dynamics and timescales of ecosystem processes; and since they can be reported as mass- or probability distributions, they provide information over a wide range in the time domain.

System age and transit time are closely related to the complexity of the ecosystem and its process rates, which are affected by the environment (Luo et al., 2017; Rasmussen et al., 2016; Sierra et al., 2017; Lu et al., 2018). In ecosystems, mean transit times of carbon seem to be much lower than mean system ages (Sierra et al., 2018b), suggesting that once a mass of carbon

enters an ecosystem, a large proportion gets quickly released back to the atmosphere, but a small proportion can remain for very long times. Furthermore, differences in transit times across ecosystems suggest that not all carbon sequestered in the terrestrial biosphere spends the same amount of time stored; e.g. one unit of phtosynthesized carbon is returned back to the atmosphere faster in a tropical than in a boreal forest (Lu et al., 2018). Therefore, not all carbon taken up from the atmosphere should be treated equally for the purpose of quantifying the climate mitigation potential of sequestering carbon in ecosystems.

Transit time distributions in ecosystems can inform us about the time newly sequestered carbon will be removed from radiative effects. This is in contrast to global warming potentials (GWPs), which quantify the radiative effects of greenhouse gases emitted to the atmosphere (Fig. 1), but do not consider the avoided radiative effect of storing carbon in ecosystems (Neubauer and Megonigal, 2015). GWPs are computed using the age distribution of $CO_2$ and other greenhouse gases in the atmosphere (Rodhe, 1990; Joos et al., 2013), but do not consider age or transit times of carbon in ecosystems for the case of

sequestration.

For more comprehensive accounting of the contribution of carbon sequestration to climate change mitigation, it is necessary to quantify the avoided warming effects of sequestered carbon in ecosystems over the timescale the carbon is stored. The GWP metric is inappropriate to quantify avoided warming potential as a result of sequestration. A metric that can capture this avoided warming effect could have applications for 1) comparing different carbon sequestration proposals considering differences in

the time carbon is stored in different proposed options, and 2) better accounting for the effect of removals by sinks in climate policy. Currently, the Intergovernmental Panel on Climate Change (IPCC) recommends countries and project developers to report only emissions by sources and removals by sinks of greenhouse gases (GHGs), treating all removals equally in terms of their fate (IPCC, 2006). In other words, all carbon taken up from the atmosphere is treated equally despite current evidence of the contrary from transit time estimations.

It has been previously recognized that GWPs have problems when applied to compute climate benefits of sequestering carbon in ecosystems (Moura Costa and Wilson, 2000; Fearnside et al., 2000; Brandão et al., 2013; Neubauer and Megonigal, 2015), and several approaches have been proposed to deal with the issue of timescales (Brandão et al., 2013). While most of these approaches deal with time as some form of delay in emissions, to our knowledge, none of them explicitly account for the time carbon is sequestered in ecosystems, since the time of carbon fixation during photosynthesis until it is returned back to

the atmosphere by respiratory processes of autotrophs and heterotrophs.

The main objective of this manuscript is to introduce a metric to assess the climate benefits of carbon sequestration in ecosystems accounting for the time carbon is stored, and to provide examples on how to use this framework for different problems of land use and ecosystem carbon management in the context of climate change mitigation. We first present the





theoretical framework for the development of the metric, and then provide three different examples for its use in ecosystem
management decisions for climate change mitigation.

## 2 Theoretical framework

### 2.1 Absolute Global Warming Potential AWGP

The direction of carbon flow, into or out of ecosystems, is of fundamental importance to understand and quantify their contribution to climate change mitigation. The absolute global warming potential (AGWP) of carbon dioxide quantifies the radiative
effects of a unit of $CO_2$ emitted to the atmosphere during its life time; in the direction land $\rightarrow$ atmosphere. It is expressed as
(Lashof and Ahuja, 1990; Rodhe, 1990)

$$\mathrm{AGWP}(T,t_0) = \int_{t_0}^{t_0+T} k_{CO_2} M_a(t)\, \mathrm{d}t \tag{1}$$

where $k_{CO_2}$ is the radiative efficiency or greenhouse effect of one unit of carbon dioxide (in mole or mass) in the atmosphere,
and $M_a(t)$ is the amount of gas remaining in the atmosphere after some time $t$ (Rodhe, 1990; Joos et al., 2013). The AGWP
quantifies the amount of warming produced by $CO_2$ while it stays in the atmosphere since the time the gas is emitted at time
$t_0$ over a time horizon $T$. The function $M_a(t)$ quantifies the fate of the emitted carbon in the atmosphere and can be written in
general form as

$$M_a(t) = h_a(t-t_0)M_a(t_0) + \int_{t_0}^{t} h_a(t-\tau)Q(\tau)\, \mathrm{d}\tau, \tag{2}$$

where $h_a(t-t_0)$ is the impulse response or Green's function of atmospheric $CO_2$; $M_a(t_0)$ is the content of atmospheric $CO_2$
at time $t_0$, and $Q(\tau)$ is the perturbation of new incoming carbon to the atmosphere between $t_0$ and $t$.

For a pulse or instantaneous emission of carbon dioxide $M_a(t_0) = E_0$, and

$$M_a(t) = h_a(t-t_0)E_0, \tag{3}$$

assuming no additional carbon is entering the atmosphere after the pulse. In case the pulse is equivalent to 1 kg or mole of
$CO_2$, $E_0 = 1$ and $M_a(t) = h_a(t-t_0)$. For a pulse emission of any arbitrary size,

$$80 \quad \mathrm{AGWP}(T,E_0,t_0) = k_{CO_2} E_0 \int_{t_0}^{t_0+T} h_a(t-t_0)\, \mathrm{d}t. \tag{4}$$

The AGWP can be computed for any other greenhouse gas using their respective radiative efficiencies and fate in the
atmosphere (Green's function). To compare different gases, the Global Warming Potential (GWP) is defined as the AGWP of a
particular gas divided by the AGWP of carbon dioxide (Shine et al., 1990; Lashof and Ahuja, 1990). Since our interest in this
manuscript is on carbon fixation and respiration in the form $CO_2$, we concentrate exclusively in AGWP.



## 2.2 Carbon sequestration CS, and the climate benefit of carbon sequestration CBS

GWPs are useful to quantify the climate impacts of increasing or reducing emissions of GHGs to the atmosphere. However, it is also necessary to quantify the climate benefits of carbon flows in the opposite direction, atmosphere → land. Furthermore, it is also important to quantify not only how much and how fast carbon enters ecosystems, but also for how long the carbon stays (Körner, 2017).

Carbon taken up from the atmosphere through the process of photosynthesis is stored in multiple ecosystem reservoirs for a particular amount of time. Carbon sequestration can be defined as the process of capture and long-term storage of $CO_2$ (Sedjo and Sohngen, 2012). We define here carbon sequestration CS over a time horizon $T$ as

$$\mathrm{CS}(T, S_0, t_0) := \int\limits_{t_0}^{t_0+T} M_s(t - t_0)\, \mathrm{d}t, \tag{5}$$

where $M_s(t - t_0)$ represents the fate of carbon in the sequestering system, and $S_0$ is the amount of fixed carbon. Notice that this definition of carbon sequestration is very similar to that of AGWP for an emission, with the exception that the radiative efficiency term is omitted.

To obtain the fate of sequestered carbon over time, we represent carbon cycling and storage in ecosystems using the theory of compartmental dynamical systems (Luo et al., 2017; Sierra et al., 2018a). In their most general form, we can write carbon cycle models as

$$\frac{\mathrm{d}x(t)}{\mathrm{d}t} = \dot{x}(t) = u(x,t) + \mathbf{B}(x,t)\, x, \tag{6}$$

where $x(t) \in \mathbb{R}^n$ is a vector of $n$ ecosystem carbon pools, $u(x,t) \in \mathbb{R}^n$ is a time-dependent vector-valued function of carbon inputs to the system, and $\mathbf{B}(x,t) \in \mathbb{R}^{n \times n}$ is a time-dependent compartmental matrix. The latter two terms can depend on the vector of states, in which case the compartmental system is considered nonlinear. In case the input vector and the compartmental matrix have fixed coefficients (no time-dependencies), the system is considered autonomous, and non-autonomous otherwise (Sierra et al., 2018a). Models expressed as an autonomous linear system have a steady-state solution given by $x^* = -\mathbf{B}^{-1} u$, where $x^*$ is a vector of steady-state contents for all ecosystem pools.

The fate of the fixed carbon can be obtained as

$$M_s(t - t_0) = \|\boldsymbol{\Phi}(t, t_0)\beta(t_0) S_0\|, \tag{7}$$

where $\beta(t_0) S_0 = u(t_0)$, and $\beta(t_0)$ is an $n$-dimension vector representing the partitioning of the total sequestered carbon among $n$ ecosystem carbon pools (Ceballos-Núñez et al., 2020). The $n \times n$ matrix $\boldsymbol{\Phi}(t, t_0)$ is the state transition operator, which represents the dynamics of how carbon moves in a system of multiple interconnected compartments (see details in appendix). Throughout this document, we use the symbol $\|\;\|$ to denote the 1-norm of a vector, i.e. the sum of the absolute values of all elements in a vector.





Because ecosystems and most reservoirs are open systems, carbon returns back to the atmosphere (mostly as ecosystem

respiration Re) according to

$$r(t) = -1^{\mathsf{T}} \mathbf{B}(t) \mathbf{\Phi}(t, t_0) \beta(t_0) S_0, \tag{8}$$

where $1^{\mathsf{T}}$ is the transpose of the $n$-dimensional vector containing only 1s.

The link between the time it takes sequestered carbon $S_0$ to appear in the output flux $r(t)$ is established by the concept of

transit time (Metzler et al., 2018). In particular, we define the forward transit time (FTT) as the age that fixed carbon will have

at the time it is released back to the atmosphere. The backward transit time (BTT) is defined as the age of the carbon in the

output flux since the time it was fixed. This implies that

$$r(t) = p_{\mathrm{BTT}}(t - t_0, t) = p_{\mathrm{FTT}}(t - t_0, t_0), \tag{9}$$

where $p_{\mathrm{BTT}}(t - t_0, t)$ is the backward transit time distribution of carbon leaving the system at time $t$ with an age $t - t_0$, while

$p_{\mathrm{FTT}}(t - t_0, t_0)$ is the forward transit time distribution of carbon entering the system at time $t_0$ and leaving with an age $t - t_0$.

For systems in equilibrium, both quantities are equal (Metzler et al., 2018). Semi-explicit formulas for these distributions are

given in the appendix.

For the atmosphere, carbon sequestration is a form of 'negative emission', and we can represent its fate in the atmosphere as

$$M_a'(t) = -h_a(t - t_0)S_0 + \int_{t_0}^{t} h_a(t - \tau) r(\tau) \, \mathrm{d}\tau, \tag{10}$$

where the prime symbol represents a perturbed atmosphere as an effect of sequestration. The first term in the rhs represents

the response of the atmosphere to an instantaneous sequestration at $t_0$, and the second term represents the perturbation in the

atmosphere of the carbon returning back from the terrestrial biosphere. Notice that the integral in this equation can be written

as a convolution $(h_a \star r)(t)$ between the impulse response function of atmospheric $CO_2$ and the respired carbon returning from

ecosystems to the atmosphere.

We define now the climate benefit of sequestration as

$$\begin{aligned}
\mathrm{CBS}(T, S_0, t_0) &:= \int_{t_0}^{t_0+T} k_{\mathrm{CO_2}} M_a'(t) \, \mathrm{d}t, \\
&= -k_{\mathrm{CO_2}} \int_{t_0}^{t_0+T} (h_a(t - t_0)S_0 - (h_a \star r)(t)) \, \mathrm{d}t.
\end{aligned} \tag{11}$$

This metric integrates over a time horizon the radiative effect avoided by sequestration of carbon in an ecosystem during the

time it is stored, and takes into account the subsequent return of the gas to the atmosphere.





## 2.3 Carbon sequestration in linear systems at equilibrium

The computation of CS and CBS is simplified for systems in equilibrium. For linear systems at steady-state, the time at which the carbon enters the ecosystem is irrelevant (Kloeden and Rasmussen, 2011; Rasmussen et al., 2016); one only needs to know for how long the carbon has been in the system to predict how much of it remains. Mathematically, this implies

$$\mathbf{\Phi}(t, t_0) = e^{a \cdot \mathbf{B}} \quad \text{for all} \quad t_0 \leq t \quad \text{and} \quad a = t - t_0. \tag{12}$$

Therefore, for linear systems at steady state, we have the special cases

$$M_s(a) = \|e^{a \cdot \mathbf{B}} u\|, \tag{13}$$

and

$$M_{s1}(a) = \left\| e^{a \cdot \mathbf{B}} \frac{u}{\|u\|} \right\|, \tag{14}$$

where $M_{s1}$ represents the fate of one unit of fixed carbon, which can also be interpreted as the proportion of carbon remaining after the time of fixation ($a = t - t_0$).

The amount of released carbon returning to the atmosphere is therefore

$$r(a) = -1^\intercal \mathbf{B} e^{a \cdot \mathbf{B}} u, \tag{15}$$

which for one unit of fixed carbon is equal to the transit time density distribution $f(\tau)$ of a linear system (Metzler and Sierra, 2018, see also appendix)

$$r_1(a) = -1^\intercal \mathbf{B} e^{a \cdot \mathbf{B}} \frac{u}{\|u\|}. \tag{16}$$

where $r_1(a) = f(\tau)$, with mean (expected value) transit time given by

$$\mathbb{E}(\tau) = -1^\intercal \mathbf{B}^{-1} \frac{u}{\|u\|} = \frac{\|x^*\|}{\|u\|}. \tag{17}$$

We can now derive the steady-state expression of CS as

$$\mathrm{CS}(T) = \int_0^T \|e^{a \cdot \mathbf{B}} u\| \, \mathrm{d}a. \tag{18}$$

Furthermore, it is possible to find a closed-form expression for this integral

$$\mathrm{CS}(T) = \|\mathbf{B}^{-1} \left( e^{T \cdot \mathbf{B}} - \mathbf{I} \right) u\|, \tag{19}$$

where $\mathbf{I} \in \mathbb{R}^{n \times n}$ is the identity matrix. Similarly, for one unit of carbon entering a steady-state system at any time, we define $\mathrm{CS}_1$ as

$$\mathrm{CS}_1(T) = \int_0^T \left\| e^{a \cdot \mathbf{B}} \frac{u}{\|u\|} \right\| \, \mathrm{d}a, \tag{20}$$





which by integration gives

$$\text{CS}_1(T) = \left\| \mathbf{B}^{-1} \left( e^{T \cdot \mathbf{B}} - \mathbf{I} \right) \frac{u}{\|u\|} \right\|. \tag{21}$$

These steady-state expressions can be very useful to compare different systems or changes to a particular system if the steady-state assumption is justified. Furthermore, it can be shown that in the long term, as the time horizon $T$ goes to infinity ($\infty$), the term $(e^{T \cdot \mathbf{B}} - \mathbf{I})$ converges to $-\mathbf{I}$, and therefore equation (19) converges to the expression

$$\lim_{T \to \infty} \text{CS}(T) = \|x^*\|, \tag{22}$$

which means that the total amount of carbon at steady-state is equal to the long-term carbon sequestration of an instantaneous amount of fixed carbon at an arbitrary time.

Similarly, for one unit of carbon entering a system at steady-state, the long-term $\text{CS}_1$ from equation (21) can be obtained simply as

$$\lim_{T \to \infty} \text{CS}_1(T) = \mathbb{E}(\tau), \tag{23}$$

by using the definition of mean transit time of equation (17). This means that long-term sequestration of one unit of $CO_2$ converges to the mean transit time of carbon in an ecosystem.

## 2.4 From instantaneous to continuous fluxes

In addition of considering isolated pulses of emissions $E_0$ or sequestrations $S_0$, we can also consider permanently ongoing emissions $e : t \mapsto E(t)$ and sequestration $s : t \mapsto S(t)$, respectively. Hence,

$$\text{CS}(T, s, t_0) := \int_{t_0}^{t_0 + T} M_s(t) \, \mathrm{d}t, \tag{24}$$

where

$$M_s(t) = \int_{t_0}^{t} \|\mathbf{\Phi}(t, \tau) \, \beta(\tau) \, s(\tau)\| \, \mathrm{d}\tau. \tag{25}$$

Here $s(\tau)$ is a scalar flux of sequestration at time $\tau$. This leads to

$$r(t) = -\mathbf{1}^{\intercal} \mathbf{B}(t) \int_{t_0}^{t} \mathbf{\Phi}(t, \tau) \, \beta(\tau) \, s(\tau) \, \mathrm{d}\tau. \tag{26}$$





The fate of sequestered carbon, for the atmosphere in the form of a balance between simultaneous sequestration and return of carbon, can now be obtained as

$$
\begin{aligned}
M_a'(t) &= -\int_{t_0}^{t} h_a(t-\tau)\,s(\tau)\,\mathrm{d}\tau + \int_{t_0}^{t} h_a(t-\tau)\,r(\tau)\,\mathrm{d}\tau \\
&= -\int_{t_0}^{t} h_a(t-\tau)\,[s(\tau)-r(\tau)]\,\mathrm{d}\tau \\
&= -(h_a \star (s-r))(t).
\end{aligned}
\tag{27}
$$

We can now define the climate benefit of sequestration as

$$
\begin{aligned}
\mathrm{CBS}(T,s,t_0) &:= \int_{t_0}^{t_0+T} k_{\mathrm{CO_2}} M_a'(t)\,\mathrm{d}t, \\
&= -k_{\mathrm{CO_2}} \int_{t_0}^{t_0+T} (h_a \star (s-r))(t)\,\mathrm{d}t.
\end{aligned}
\tag{28}
$$

This expression of CBS accounts for the dynamic behavior of inputs and outputs of carbon in ecosystems, and can be used to represent time-dependencies resulting from disturbances, or produced by emission scenarios or scheduled management activities. This time-dependent CBS is computed for a time horizon $T$ starting at any initial time $t_0$. In other words, it can be used to analyze specific time windows of interest.

## 2.5   The radiative efficiency $k_{\mathrm{CO_2}}$ of carbon dioxide and its impulse response function $h_a(t)$

The radiative efficiency of carbon dioxide is a function of the concentration of this gas and the concentration of other gases in the atmosphere with overlapping absorption bands (Lashof and Ahuja, 1990; Shine et al., 1990). Therefore, $k_{\mathrm{CO_2}}$ changes as the concentration of GHGs change in the atmosphere. Time-dependent radiative efficiencies $k_g(t)$ of a gas $g$ can be considered in the calculation of CBS, which would imply a numerical integration of the time-dependent integrals presented above. For most applications however, the radiative efficiency of $CO_2$ has been assumed constant in the limit of a small perturbation at a specific background concentration (Lashof and Ahuja, 1990; Shine et al., 1990; Joos et al., 2013; Myhre et al., 2013).

Here, we use a constant value of $k_{\mathrm{CO_2}} = 6.48 \times 10^{-3}$ W m$^{-2}$ PgC$^{-1}$ based on results reported by Joos et al. (2013) for an atmospheric background of 389 ppm ($\sim$ present day). This radiative efficiency represents the change in radiative forcing caused by a change of 1 Pg of carbon in the atmosphere in the form of carbon dioxide in units of rate of energy transfer (Watt) per square meter of surface.

Joos et al. (2013) have also derived impulse response functions of carbon dioxide in the atmosphere using coupled carbon-climate models (Figure 2). One function was derived for a pre-industrial atmosphere with 280 ppm and another for a present day atmosphere with 389 ppm. The functions they report are averages for multiple models fitted to a sum of exponential functions that include an intercept term. This intercept implies that a proportion of the added carbon dioxide never leaves from the



atmosphere-ocean-terrestrial system to long-term geological reservoirs. An alternative function was proposed by Lashof and
Ahuja (1990) that omits the intercept term (Figure 2). We evaluate these different functions for the purpose of this manuscript.

## 3 Example 1: the climate benefit of carbon sequestration in the pre-industrial biosphere

We will show now an application of the theory using a simple global carbon model to compute CS and CBS assuming a
terrestrial biosphere in steady-state. The model was initially developed by Emanuel et al. (1981) and contains five main com-
partments: non-woody tree parts $x_1$, woody tree parts $x_2$, ground vegetation $x_3$, detritus/decomposers $x_4$, and active soil
carbon $x_5$ (Figure 3). In addition to its simplicity and tractability, there are two advantages of using this model over others: 1)
it provides reasonable values of carbon stocks and fluxes for a pre-industrial biosphere, 2) its impulse response function and
distributions of system age and transit time have been studied previously (Emanuel et al., 1981; Thompson and Randerson,
1999; Metzler and Sierra, 2018).

The model, expressed as a linear autonomous compartmental system, is given by

$$\dot{x} = u + \mathbf{B}x,$$

$$= \begin{pmatrix} 77 \\ 0 \\ 36 \\ 0 \\ 0 \end{pmatrix} + \begin{pmatrix} -77/37 & 0 & 0 & 0 & 0 \\ 31/37 & -31/452 & 0 & 0 & 0 \\ 0 & 0 & -36/69 & 0 & 0 \\ 21/37 & 15/452 & 12/69 & -48/81 & 0 \\ 0 & 2/452 & 6/69 & 3/81 & -11/1121 \end{pmatrix} \begin{pmatrix} x_1 \\ x_2 \\ x_3 \\ x_4 \\ x_5 \end{pmatrix}, \tag{29}$$

where mass of carbon is in units of PgC, and fluxes in units of PgC yr$^{-1}$. Total carbon inputs to the terrestrial biosphere (gross
primary production GPP) are 113 PgC yr$^{-1}$.

Applying equations (13) and (14), it is possible to observe the fate of the total incoming carbon entering an arbitrary time
(Figure 4). Carbon enters the terrestrial biosphere through the non-woody vegetation (leaves) and ground vegetation pools. A
large proportion of the carbon that enters at any given time is quickly respired or transferred from these pools to the woody tree
parts, detritus and soil. After some decades, most of the remaining carbon is transferred to the active soil carbon compartment
where it is eventually respired by microorganisms and returned back to the atmosphere with a mean transit time of 15.1 yr for
the whole system. Half of the sequestered carbon is returned back to the atmosphere in 2.3 yr, and 95% in 74.5 yr.

Carbon sequestration, i.e. the area under the curve of the amount of remaining carbon over time, shows an increasing and
asymptotic behavior as the time horizon of integration increases (Figure 5a). Here, CS is reported in units of PgC yr, because
this is the amount of carbon retained over a fixed time horizon. For time horizons of 50, 100, 500, and 1000 yr, CS was
1012.19, 1300.83, 1704.49, and 1711.22 PgC yr, respectively. In the long-term (as the time horizon goes to infinity), carbon
sequestration converges to the steady-state carbon stock predicted by the model of 1711.3 PgC.





A similar computation can be made for one unit of fixed carbon (unitless). In this case $CS_1$ was 8.96, 11.51, 15.08, and 15.14 yr for time horizons of 50, 100, 500, and 1000 yr, respectively. In the long-term, $CS_1$ converges to the mean transit time of carbon, 15.14 yr (Figure 5b).

Due to sequestration at $t_0$, the CBS shows a rapid negative increase in radiative forcing, which decreases as the time horizon increases due to the return of carbon to the atmosphere as an effect of respiration (Figure 5c). The shape of the curve however, depends strongly on the impulse response function for atmospheric $CO_2$. CBS is larger over the long-term ($> 300$ yr) for the PI100 curve proposed by Joos et al. (2013) due to its intercept. This means that emitted carbon that never leaves the atmosphere-ocean-terrestrial system can be retained in the biosphere with a fixed benefit over an infinite time horizon. This is in contrast to the impulse response function proposed by Lashof and Ahuja (1990) in which emitted carbon eventually leaves the atmosphere-ocean-terrestrial system and returns to a geological reservoir. The CBS in this case returns to zero in the long-term, which means that the climate benefit of sequestration is temporary and does not last forever. We believe this latter case is more realistic, and therefore we will use the impulse response function of Lashof and Ahuja (1990) from here on.

The AGWP computed with both impulse response functions predicts very large impacts of carbon emissions compared to sequestration (Figure 5d). Over a 100 year time horizon, the AGWP is above 45 W m$^{-2}$ yr; and over a 1000 year time horizon, AGWP is above 170 W m$^{-2}$ yr. In contrast, CBS values are never larger (in absolute value) than 4 W m$^{-2}$ yr, suggesting that the effects of emitting 1 PgC to the atmosphere are much more dramatic than the avoided warming produce by the sequestration of 113 PgC in the pre-industrial biosphere.

## 4   Carbon management to maximize the climate benefit of carbon sequestration

In the context of climate change mitigation, management of ecosystems may be oriented to increase carbon sequestration and its climate benefit. In the recent past, scientists and policy makers have advocated increasing the amount of inputs to ecosystems as an effective form of carbon management (e.g. Silver et al., 2000; Grace, 2004; Lal, 2004; Chabbi et al., 2017; Minasny et al., 2017). Although increases in carbon inputs can increase the amount of stored carbon in an ecosystem with related climate benefits, it does not necessarily increase the amount of time the sequestered carbon will stay in the system. Therefore, strategies that focus on increasing carbon inputs alone, do not take full advantage of the potential of ecosystems to mitigate climate change.

We can conceptualize any management activity that increases or reduces carbon inputs to an ecosystem by a factor $\gamma$, so the new inputs are given by the product $\gamma u$. For example, if we increase carbon inputs to an ecosystem by 10%, $\gamma = 1.1$. Increasing (decreasing) carbon inputs increase (decrease) carbon storage at steady state by an equal proportion since

$$
\begin{aligned}
-\mathbf{B}^{-1}(\gamma u) &= \gamma(-\mathbf{B}^{-1}u), \\
&= \gamma x^*.
\end{aligned}
\tag{30}
$$





However, the time carbon requires to travel through the ecosystem is still the same since the transit time does not change, as we can see from the mean transit time expression

$$-1^\intercal \mathbf{B}^{-1} \frac{\gamma u}{\|\gamma u\|} = \mathbb{E}(\tau). \tag{31}$$

Both the transit time distribution (eq. B4) and the mean transit time (eq. 17) only take into account the proportional distribution of the carbon inputs to the different pools $(u/\|u\|)$, but not the total amount of inputs. Therefore, a unit of carbon that enters 270 an ecosystem stays there for the same amount of time independent of how much carbon is entering the system. Although these results only apply to linear systems at steady-state, they provide some intuition about what might be the case in systems out of equilibrium, such as while in transition from one steady state to another following a land use or management change.

Carbon management can also be oriented to modify process rates in ecosystems as encoded in the matrix $\mathbf{B}$. A proportional decrease (increase) in process rates by a factor $\xi$ not only increases (decreases) carbon storage as

$$
\begin{aligned}
\quad -(\xi \mathbf{B})^{-1} u &= \frac{1}{\xi}\left(-\mathbf{B}^{-1} u\right), \\
&= \frac{x^*}{\xi}, \tag{32}
\end{aligned}
$$

it also increases (decreases) the mean transit time as

$$-1^\intercal (\xi \mathbf{B})^{-1} \frac{u}{\|u\|} = \frac{\mathbb{E}(\tau)}{\xi}. \tag{33}$$

Based on these results, it is now clear that carbon management to increase carbon inputs alone can only increase CS, but not 280 $CS_1$; i.e. the new carbon inputs have a sequestration benefit only through increase of carbon storage but not through a longer transit time in ecosystems. Management to decrease process rates on the contrary, can increase both CS and $CS_1$ because the new carbon entering the system stays there for longer.

We can see these effects of carbon management on CS by running simulations using the model of Example 1 (Figure 6). In this example, we modified carbon inputs and process rates by either increasing them by 10 and 50% ($\gamma$, $\xi$ = 1.1, 1.5), or 285 decreasing them by 10 and 50% ($\gamma$, $\xi$ = 0.9, 0.5). The simulations showed that increasing or decreasing carbon inputs increase or decrease CS for any time horizon (Figure 6a), but it does not modify the behavior of one unit of sequestered carbon ($CS_1$) (Figure 6b). On the contrary, decreasing or increasing process rates increase or decrease both CS (Figure 6c) and $CS_1$ (Figure 6d).

Management of inputs or process rates have subsequent effects on CBS. If the amount of sequestered carbon is larger than 290 a reference system, the climate benefit is larger and can last for a longer time horizon (Figure 7). Similarly, if process rates decrease with respect to a reference system, the CBS is larger and last for a longer time horizon. Therefore, a combination of management of carbon inputs and process rates can have large benefits for climate change mitigation.

These results imply that carbon management can be planned to maximize the climate benefits of carbon sequestration by both increasing carbon inputs and increasing transit time. Examples on how to increase transit time in ecosystems are discussed 295 in subsequent examples.



## 5 CS and CBS during the industrial period

During the industrial period, starting in calendar year 1850, concentrations of carbon dioxide in the atmosphere increased steadily with a number of consequences on Earth system processes. Models participating in the Coupled Model Intercomparison Project Phase 6 (CMIP6) (Eyring et al., 2016), predict an increase in carbon uptake by the terrestrial biosphere, i.e. gross primary production, from a multi-model average of 133.6 PgC yr$^{-1}$ in 1850 to 160.8 PgC yr$^{-1}$ in 2014 (Figure 8a). To demonstrate the application of our framework for the out of steady-state case, we use here this time series of average GPP to drive the model of Emanuel et al. (1981) during the industrial period.

Carbon incorporated into the terrestrial biosphere returns back to the atmosphere with an average transit time of 15.1 yr; therefore a large proportion of carbon incorporated during the industrial period is respired back quickly while some carbon is stored for a longer time (Fig. 8a). Using equation (25), we can compute the integrated fate of carbon incorporated every year $M_s(t)$, from 1850 to any other year until 2014 (Fig. 8b), which is the total amount of all remaining carbon since the beginning of the industrial period in $t_0 = 1850$. This amount of carbon can be integrated further for different time horizons $T$, using equation (24) to obtain time-dependent values of CS (Fig. 8c); i.e. for different lengths of the time horizon $T$.

Starting in the year 1850 and for time horizons of 50, 100, and 150 yr, the predicted values of CS are 40,732, 110,811, and 196,569 PgC yr, respectively (Fig. 8b). As the time horizon increases, CS increases due to the continuous accumulation of extra carbon in the terrestrial biosphere (Fig. 8c). This upward trend is not surprising because even if the system were at steady-state, the continuous accumulation of annual fluxes would lead to a linear trend with time horizon (dashed line in Fig. 8c).

As the time horizon increases and more carbon is retained, the climate benefit of sequestration increases (negative increase)(Fig. 8d). This prediction of CBS for the industrial period represents the total cumulative amount of avoided warming prevented by carbon sequestered in the terrestrial biosphere. It can be computed for different time horizons and initial times $t_0$, and could be useful to compare with values of AGWP for GHG emissions or with different forms of biospheric management.

## 6 Example 2: Carbon management in forests

Land use and land use change are important drivers of changes in the terrestrial carbon cycle, with forest-related activities considered as a major player in the global carbon balance. Over the years, there has been considerable debate about how to quantify the role of forestry in mitigating climate change (e.g. Harmon et al., 1990; Fearnside, 1995; Tipper and de Jong, 1998; Winjum et al., 1998; Brown et al., 1999; Niles and Schwarze, 2001; Nabuurs and Sikkema, 2001; Werner et al., 2010; Körner, 2017; Luyssaert et al., 2018; Schulze et al., 2019; Sato and Nojiri, 2019, among others). One contended issue is how to quantify the contribution of forest products with relatively long lifetimes that can help to lockup carbon. How to account for time and permanence of sequestered carbon has been another contended issue (Fearnside et al., 2000; Moura Costa and Wilson, 2000; Brandão et al., 2013) as well as quantifying the climate benefits of bioenergy as a substitute of fossil-fuel derived energy (Schulze et al., 2019). The definition of carbon sequestration we introduced here, and the CBS concept we developed, can help to better address these issues.





**Table 1.** Additional pools included in the model to represent the fate of harvested wood from the tree woody pool. $B_i$ represents the cycling rate of each pool in units of $\text{yr}^{-1}$. Harvest distributions represent the proportion of the harvested wood that is transferred to each pool.

|  | $B_i$ | Harvest distribution, S1 | Harvest distribution, S2 and S3 |
|---|---|---|---|
| Long duration $x_6$ | 0.02 | 0.60 | 0.30 |
| Mid duration $x_7$ | 0.04 | 0.20 | 0.20 |
| Short duration $x_8$ | 0.32 | 0.10 | 0.10 |
| Bioenergy $x_9$ | 0.70 | 0.05 | 0.30 |
| Waste $x_{10}$ | 1.00 | 0.05 | 0.10 |

Commercial forest management is commonly advocated as an activity that can play a major role in climate change mitigation as opposed to preserving old-growth forests. The rationale is that young managed forests have generally a larger rate of biomass growth than old-growth forests, and in addition, wood products can retain carbon for a considerable amount of time (Schulze et al., 2019). Arguments against commercial forestry highlight that total carbon stocks are lower in younger than in old-growth forests, which results in a long-term net loss of carbon to the atmosphere. Also, inefficiencies in the wood production chain result in significant carbon losses due to waste with only minor proportions of harvested wood ending up in long-duration products (Harmon et al., 1990).

In this example, we will not advocate any of the fore-mentioned points of view, but will show that the CS and CBS concepts can be an important tool to address this debate. For the sake of simplicity, let's consider the model of Emanuel et al. (1981) again, and introduce a flow of carbon from the woody tree parts pool $x_2$ to a set of five new pools, namely: long- $x_6$, mid- $x_7$, and short-duration $x_8$ forest products, as well as a bioenergy $x_9$ and a waste pool $x_{10}$ (Figure 3). Carbon in each of these pools have a distinct cycling rate according to Table 1. This choice of values is somewhat arbitrary, and they may change according to tree species, silvicultural practices, and climatic factors. Other transfers among these and other compartments (e.g. landfill disposal) are also ignored in this simple example.

We will consider three different cases of carbon management with different silvicultural practices and efficiencies in terms of the amount of carbon that ends up in long-duration forest products. In scenario S1, 60% of the harvested wood is transferred to long-duration products and only 5% ends up in waste, and 5% is used in bioenergy. Carbon in these two last pools is released quickly to the atmosphere. In scenario S2, only 30% of the harvested wood is transferred to long-duration products and much larger proportions, 30 and 10%, end up in bioenergy and waste, respectively (Table 1). In scenarios S1 and S2, silvicultural practices are such that they result in an increase in the flow from the non-woody tree parts to the detritus pool by 30% in comparison to the original model; i.e. an additional flux of 11 PgC $\text{yr}^{-1}$ among these pools. In scenario S3, we considered improved silvicultural practices that decrease the transfer of non-woody tree parts to detritus by 10 PgC $\text{yr}^{-1}$, and instead, this amount is transferred to the woody-tree parts. The distribution of the harvested wood in S3 is similar as in S2 (Table 1), which implies that S3 is a scenario of improved silvicultural practices but low efficiency in the transformation of harvested wood.





One main consequence of transferring a large amount of carbon from wood biomass to forest products is that the steady-state ecosystem carbon stock is lower by 588.5 PgC in the S1 and S2 scenarios, and by 652.0 PgC in the S3 scenario, compared to the no management case. For S1 and S2, this is 34% less carbon compared to the original amount of carbon in the no management case, and 38% less carbon in S3. However, if we include the amount of carbon stored in the forest products, the steady-state carbon stock is 187.4 PgC higher in the high efficiency transfer scenario S1, 11% higher with respect to the original model. In the low efficiency transfer scenario S2, the steady-state carbon stock is lower by 131 PgC (8% less) with respect to the no management case. In S3, improved silvicultural practices compensate losses due to low efficiencies in transformations to long-duration products, resulting in 39.5 PgC more (23%) than in the no management case.

Due to carbon transfers to wood products, the amount of carbon that enters the ecosystem in any given year decreases faster in all three scenarios during the first decades in comparison with the no management case (Figure 9a,b). In subsequent decades, due to transfers to wood products, carbon is lost to the atmosphere more slowly with differences among the scenarios according to transfer efficiencies and silvicultural management. In the long term ($> 200$ yr), all scenarios converge, which implies that in all cases the carbon that enters in a particular year is eventually returned back to the atmosphere (Figure 9a,b).

The CS concept helps to disentangle the contrasting effects of fast losses in the initial years after carbon enters versus the slow loss in later decades because it integrates the mass loss curves (Figure 9c). Management practices that can retain carbon for a longer time result in higher amounts of carbon sequestration in the long-term. If we subtract the CS computed for the original model with no management from the CS computed for the three scenarios, we see that scenarios S1 and S3 result in an increase in carbon sequestration, while the low efficiency transfer scenario S2 results in a decrease (Figure 9d). Low efficiencies in carbon transfers to long-duration products can be compensated by silvicultural practices that reduce the amount of detritus and increase transfers to wood as shown by the CS of the S3 scenario (Figure 9d). However, if the CS were computed for short time horizons ($< 30$ yr), the CS difference would be negative in all cases due to the fast losses of carbon in the initial decades.

These differences in the fate of carbon and CS lead to similar qualitative differences in terms of CBS (Figure 9e,f). In all cases, CBS is large after the instantaneous uptake and differences among scenarios spread as the time horizon increases depending on how carbon is transferred among the different reservoirs and is returned back to the atmosphere. In the long-term, all scenarios converge to zero, which implies that the benefits of sequestering a given amount of carbon at a particular time decline over large time horizons.

These results suggest that forest management for climate change mitigation may be a viable alternative if it leads to a considerable increase in transit time; i.e., if a relatively large proportion of harvested wood is transferred to long-duration products with small losses to waste, and silvicultural practices are adopted that reduce the amount of detritus and increase allocation of C to wood.

As mentioned before, another contended issue related to forestry projects for climate change mitigation is how to quantify potential benefits due to fuel substitution by bioenergy (Werner et al., 2010; Schulze et al., 2019). The CBS can be computed for individual pools if the curve of mass of remaining carbon for the pool is known (Figure 10). Since the CBS represents the quantity of avoided warming in units of W m$^{-2}$ yr, it can be easily compared to the AGWP of any amount of fossil fuel emitted to the atmosphere that corresponds to the substitution. The fact that the CBS and the AGWP share the same units makes such





comparisons and computations straightforward, which can be used in integrated assessments that also consider the quantity of fossil fuels required to manage and harvest forests.

For the three scenarios considered in this example, the CBS is larger for the bioenergy pool in the S3 scenario because more

carbon is allocated to wood and more of the harvested wood is transferred to the bioenergy pool in comparison to the S1 and S2 case (Figure 10). In the long-term, the CBS for the bioenergy pool can be simply computed as $k_{CO_2} x_9^*$ following equation (22) applied to individual pools. Therefore, quantifying the long-term contribution of bioenergy as a climate benefit can be an easy calculation if the steady-state assumption can be justified.

The role of forest ecosystems in global climate however, goes beyond the effects related to the carbon cycle. Forests also

influence climate by effects on albedo, partitioning of latent and sensible heat fluxes, and effects on land surfaces roughness (Bonan, 2008). These effects can be quantified as radiative effects in energy units (e.g. W m$^{-2}$), therefore the CBS provides information in units more comparable to those used to assess the overall effect of forests on climate.

## 7 Example 3: climate benefit of carbon sequestration in soils

In addition to carbon management in forests and wood products, soils are commonly considered as a promising alternative

to sequester carbon and mitigate climate change (Lal, 2004; Minasny et al., 2017). Carbon in soils can be stabilized by a variety of physicochemical and biological mechanisms that can considerably prolong the time carbon stays in the soil system. Mechanisms for carbon stabilization and destabilization in soils interact in multiple forms (Sollins et al., 1996; von Lützow et al., 2006; Dungait et al., 2012), which results in a large heterogeneity of process rates and therefore in the transit time of carbon (Sierra et al., 2018b). Mean transit times of soil carbon can vary from a few years to centuries as predicted by different

global-scale soil carbon models (Luo et al., 2017; Sierra et al., 2018b; Lu et al., 2018). This is because once organic matter enters the soil in the form of plant detritus, it gets quickly consumed by microorganisms, most of it gets quickly respired and emitted to the atmosphere in the form of $CO_2$, but a small proportion can be transformed to different chemical forms and can also get sorbed into mineral surfaces where it can be retained for centuries to millennia (Trumbore, 2009; Rasmussen et al., 2018). Understanding the fate of carbon in soils, particularly in the long-term, is of fundamental importance to better

understand the climate benefits of carbon sequestration in terrestrial ecosystems.

The model of Emanuel et al. (1981) that we have considered so far, contains a simple representation of soil carbon in a homogeneous active pool. To better explore the fate of carbon in a heterogeneous soil and how this representation affects the CBS, we will now replace the active pool of this model with the structure of the well-known Century model (Parton et al., 1987), which contains an active, a slow, and a passive pool, with sequential transfers of carbon among them (Figure 3). Century

is the basis of many Earth system models, and although it has been criticized for not including detailed processes, it still has been useful in many different studies to predict long-term soil carbon storage (Blankinship et al., 2018). The implementation of the model included here follows the description of the original model in Parton et al. (1987), with default parameters not modified by temperature, moisture, or soil texture.





The inclusion of a heterogeneous soil structure results in a more rapid release of carbon in the first decades after carbon
enters the system in comparison to the original model (Figure 11a). However, after the first 70 years more carbon stays in the
system and it is lost at a much slower rate in comparison to the original model. The reason for this behavior is that the active
pool in Century cycles much faster than the aggregated active pool of the original model. The slow and passive pools on the
contrary, cycle carbon at much slower rates and therefore carbon is retained in the system for a much longer time.

The long-term CS of an annual input of 113 PgC in the model with soil-pool structure according to Century is much larger
than that of the original model and all other forest management scenarios considered in the previous example (Figure 11b).
However, at short-time scales ($< 150$ yr) the CS of the extended model is lower than in all other cases, except for S2. This
means that if we compare carbon sequestration in wood products versus soil carbon, the climate benefit would depend strongly
on the time horizon of integration, and one can obtain contrasting answers depending on this integration time. More generally,
the integration time in the computation of CS must capture as much as possible the range of timescales of carbon cycling in a
system. If the integration time is too short and does not capture long-term slow processes, it would underestimate the long-term
CS.

Similarly, the CBS in the system with soil structure according to Century is much larger than the CBS in the other manage-
ment cases considered before and can last for much longer (Figure 11c). Since the CBS in these examples is computed for the
same amount of carbon uptake (113 PgC), we can see the importance of long transit times for climate change mitigation. The
mean transit time of the original model of Emanuel et al. (1981) is 15.1 yr, and for the forest management scenarios S1, S2, and
S3, it is 16.8, 14.0, and 15.5 yr, respectively. For the expanded model with soil-pool structure according to Century, the mean
transit time is 49.5 yr. We can thus see that the large value of CBS for this last case is due to a longer transit time of carbon in
the ecosystem.

Soils are indeed a promising reservoir to store carbon in ecosystems, and the climate benefits of sequestering carbon in soils
may be larger than the climate benefits of other forms of ecosystem management. However, our steady-state assumption implies
that changes in management in soils must be sustained for very long times for them to be relevant. It becomes then a practical
challenge to promote sustained carbon sequestration in soils over centuries (Amundson and Biardeau, 2018; Schlesinger and
Amundson, 2019).

## 8 Discussion

Debates about the role of ecosystems in climate change mitigation have given disproportionate attention to quantifying sources
and sinks of carbon, but much less attention to the fate of carbon once it enters an ecosystem. The time carbon remains stored in
an ecosystem is relevant for climate change mitigation because during this time the carbon is removed from radiative effects in
the atmosphere. In this manuscript, we propose a metric that can integrate both the amount of carbon that enters the ecosystem
and the time it is stored there while avoiding radiative effects: the climate benefit of sequestration (CBS). This metric is strongly
controlled by both the amount of carbon inputs to an ecosystem and the aggregated effect of all process rates at which this





carbon is cycled before getting released back to the atmosphere. This aggregated effect can be encapsulated in the concept of transit time, which determines long-term carbon sequestration CS.

The CBS concept builds on the concept of absolute global warming potential (AGWP) of a greenhouse gas, with the main difference that CBS quantifies avoided warming during the time carbon is stored in an ecosystem, while AGWP quantifies
potential warming when the carbon enters the atmosphere. Both metrics rely on the quantification of the fate of carbon (or other GHGs for AGWP) once it enters the system. For atmospheric systems, a significant amount of work has been done in determining the fate of GHGs after emissions (e.g. Rodhe, 1990; O'Neill et al., 1994; Prather, 1996; Archer et al., 2009; Joos et al., 2013). For terrestrial ecosystems however, robust methods to quantify the fate of sequestered carbon have been developed only recently (Rasmussen et al., 2016; Metzler and Sierra, 2018; Metzler et al., 2018).

Other concepts have been proposed in the past to account for the time carbon is stored in ecosystems (see review by Brandão et al., 2013, and references therein), with special interest in accounting for carbon credits. None of these concepts explicitly account for the time carbon is retained in the ecosystem, but rather use concepts related to delay in emissions (Fearnside et al., 2000) or equivalence of carbon storage in relation to AGWP (Moura Costa and Wilson, 2000). One notable exception are the concepts of sustained global warming potential SGWP and sustained global cooling potential SGCP proposed by Neubauer
and Megonigal (2015). Our CBS concept captures some of the ideas of the SGCP concept, but differs in some fundamental assumptions related to the interpretation of Green's functions, the treatment of time-dependent fluxes and rates, and reporting. While SGCP reports values in reference to $CO_2$ as is commonly done for GWP, we report CBS for individual gases as it is done for AGWP. Appendix A elaborates on other aspects of the SGWP and SGCP concepts.

Compared to previously proposed metrics, the concept of CBS can be very useful to address some of the existing debates
about the role of ecosystems in mitigating climate change. For example, it can be used to better account for the climate impacts of storing carbon in long-term reservoirs and the climate benefits of increasing the transit time in these systems. It can also be used to better quantify the role of bioenergy as fuel substitution by computing the CBS of the bioenergy pool and adding the negative AGWP caused by the avoided emission. Similarly, it can be incorporated in assessments of sequestration in industrial systems with associated carbon capture and storage. Furthermore, it can be combined with quantifications of AGWP that assess
the climate impact of emissions to obtain assessments of the net climate effect of an ecosystem or some management.

The global warming potential (GWP) is commonly used to quantify the climate impact of an emission of a certain gas in relation to the impact of an emission of $CO_2$. Projects on avoided deforestation, land use change, and even enhanced carbon sequestration have relied on this metric to assess climate benefits or impacts. However, this metric has two limitations in comparison to the combined use of CBS and AGWP we advocate here: 1) it only quantifies the climate effects of emissions
but not of sequestration, and treats all fixed carbon equally independent of its transit time in the ecosystem, 2) it is a relative measure with respect to the emission of $CO_2$. Therefore, GWPs are reported in units of $CO_2$-equivalents, which only address indirectly the effect of a gas in producing warming. The CBS on the contrary, quantifies the effects of avoided warming in units of W m$^{-2}$ times the amount of time an amount of carbon is retained, which can be better compared to other effects of ecosystems on climate (Bonan, 2008).



Carbon management in ecosystems can be targeted to maximize CS and CBS, which can be achieved by not only increasing carbon inputs, but also by increasing the transit time of carbon. There are many ways in which the transit time of carbon can be increased; for instance, by increasing transfers of carbon to slow cycling pools such as the case of increasing wood harvest allocation to long-duration products (Schulze et al., 2019) or addition of biochar to soils, or by reducing cycling rates of organic matter such as the case of soil flipping (Schiedung et al., 2019). Independently of the management activity, CS and CBS can
be powerful metrics to quantify their climate benefits, make comparisons among them, and compare against baselines or no management cases.

    The examples we provided in this manuscript are only for illustration purposes of the CBS concept, and by no means we advocate any of the management activities discussed here before they are studied more carefully. For more precise quantifications of the CBS, more detailed and reliable models should be used. The model of Emanuel et al. (1981) is an excellent
tool to illustrate carbon cycle related concepts because of its simplicity and tractability, but other models with more accurate parameterizations and including more processes should be considered for practical applications. In any case, the computation of the CBS relies on a model, which can be as simple as a one-pool model or a state-of-the-science land surface model. For simplicity, we relied on the steady-state assumption in some of our examples, but the formulas and formal theory developed in Section 2 are general enough to deal with the non-steady-state case as well as with models with nonlinear interactions among
state variables.

    Two potential limitations to apply the concepts of CS and CBS are that 1) they rely on a model that tracks the fate of the fixed carbon and 2) on a Green's or impulse response function of $CO_2$ in the atmosphere. Reliable models may not be available for certain type of ecosystems or may include large uncertainties that propagate to CS and CBS estimates. Also, estimates of Green's functions for atmospheric $CO_2$ seem to have also large uncertainties, particularly for long timescales
(Archer et al., 2009; Lashof and Ahuja, 1990). The functions proposed by Joos et al. (2013), derived from complex simulations of coupled climate-carbon models, produce unrealistic behaviors in the long-term due to the infinite storage of an emission in the atmosphere-ocean-terrestrial system. They are also derived from models out of equilibrium, violating the steady-state and linearity assumptions of the linear response theory. Advances in our understanding of the fate of emitted $CO_2$ in the atmosphere will consequently derive in better estimates of the climate benefits of carbon sequestration.

## 510  9  Conclusions

    Analyses of carbon sequestration for climate change mitigation purposes must consider both the amount of carbon inputs and the transit time of carbon. Both concepts are encapsulated in the concepts of carbon sequestration CS and climate benefit of sequestration CBS that we propose. Carbon management can be oriented to maximize CS and CBS, which can be achieved by managing both rates of carbon input and process rates in ecosystems. We believe the use of these metrics can help to better
deal with current discussions about the role of ecosystems in mitigating climate change, and will provide better estimates of avoided or human-induced warming.





*Code availability.* Code to reproduce all results is available at https://git.bgc-jena.mpg.de/csierra/cbs. Upon acceptance of the manuscript, a copy of this repository will be archived in a permanent location with a respective digital object identifier.

## Appendix A: Comment on Neubauer and Megonigal (2015)

Neubauer and Megonigal (2015) proposed two metrics, the sustained global warming potential SGWP and the sustained global cooling potential SGCP, to overcome issues with GWP. However, there is an important misconception in their study that we would like to address here. In particular, these authors state " . . . *GWPs requires the implicit assumption that greenhouse gas emissions occur as a single pulse; this assumption is rarely justified in ecosystem studies*". The use of pulse emissions in computing AGWP, as shown in equation (3), is done with the purpose of obtaining a representation of the fate of a unit of

emissions under the assumption that the system is in equilibrium. This is a mathematical property of linear time-invariant dynamical systems, by which an impulse response function can provide a full characterization of the dynamics of the system (Hespanha, 2009). In other words, the emission pulse is a mathematical method to obtain a description of the fate of incoming mass into the system, but it is not an assumption required on the system.

To use impulse response functions, it is necessary to assume that a system is in equilibrium and that all rates remain constant

for all times. It is this assumption that is problematic and difficult to impose on ecosystems, but not the pulse emission because it is simply a method. Therefore, we are of the opinion that the sustained-flux global warming potential metric proposed by these authors is unjustified on the argument that it removes the assumption of pulse emissions.

One interesting characteristic of the study of Neubauer and Megonigal (2015) is that it uses a model that couples an ecosystem compartment with the atmosphere, and their computation of SGWP and SGCP captures the interactions between these two

reservoirs similarly as in the framework described here in section 2. The SGCP is very similar in spirit to the CBS. However, their approach differs from the approach we present here in that our mathematical framework is general enough to deal with ecosystem models of any level of complexity, not restricted to a one pool model and constant parameters and sequestration rates. Furthermore, we abstain from proposing a metric that is relative to $CO_2$. We are rather interested in an absolute metric that quantifies the effect of $CO_2$ sequestration on radiative forcing, and not in equivalents to sequestration or emissions of other

gases.

## Appendix B: Fate and timescales of carbon in compartmental systems

Carbon cycling in the terrestrial biosphere is well characterized by a particular type of dynamical systems called *compartmental systems* (Anderson, 1983; Jacquez and Simon, 1993). These systems of differential equations generalize mass-balanced models and therefore generalize element and carbon cycling models in ecosystems (Rasmussen et al., 2016; Luo et al., 2017; Sierra

et al., 2018a). In their most general form, we can write carbon cycle models as

$$\frac{\mathrm{d}x(t)}{\mathrm{d}t} = \dot{x}(t) = u(x,t) + \mathbf{B}(x,t)\,x, \tag{B1}$$





where $x(t) \in \mathbb{R}^n$ is a vector of ecosystem carbon pools, $u(x,t) \in \mathbb{R}^n$ is a time-dependent vector-valued function of carbon inputs to the system, and $\mathbf{B}(x,t) \in \mathbb{R}^{n \times n}$ is a time-dependent compartmental matrix. The latter two terms can depend on the vector of states, in which case the compartmental system is considered nonlinear. In case the input vector and the compartmental matrix have fixed coefficients (no time-dependencies), the system is considered autonomous, and non-autonomous otherwise (Sierra et al., 2018a). At steady-state, the autonomous linear system has the general solution $x^* = -\mathbf{B}^{-1} u$.

The probability density function (pdf) for system age of linear autonomous models at steady-state can be computed by the following expression (Metzler and Sierra, 2018)

$$f(a) = -1^\intercal \mathbf{B} \, e^{a \cdot \mathbf{B}} \, \frac{x^*}{\|x^*\|}, \quad a \geq 0, \tag{B2}$$

where $a$ is the random variable age, $1^\intercal$ is the transpose of the $n$-dimensional vector containing ones, $e^{a \cdot \mathbf{B}}$ is the matrix exponential computed for each value of $a$, and $\|x^*\|$ is the sum of the stocks of all pools at steady-state.

The mean, i.e. the expected value, of the age pdf can be computed by the expression

$$\mathbb{E}(a) = -1^\intercal \mathbf{B}^{-1} \frac{x^*}{\|x^*\|} = \frac{\|\mathbf{B}^{-1} x^*\|}{\|x^*\|}. \tag{B3}$$

The pdf of the transit time variable $\tau$ for linear autonomous systems in equilibrium is given by (Metzler and Sierra, 2018)

$$f(\tau) = -1^\intercal \mathbf{B} \, e^{\tau \cdot \mathbf{B}} \, \frac{u}{\|u\|}, \quad \tau \geq 0, \tag{B4}$$

and the mean transit time by

$$\mathbb{E}(\tau) = -1^\intercal \mathbf{B}^{-1} \frac{u}{\|u\|} = \frac{\|x^*\|}{\|u\|}. \tag{B5}$$

For the most general case of nonlinear non-autonomous systems, we follow the approach described in Metzler et al. (2018). For these systems, the age distribution of mass is given by

$$\text{Mass in the system at time } t \text{ with age } a = \begin{cases} \mathbf{\Phi}(t, t-a) \, u(t-a), & a < t - t_0, \\ \mathbf{\Phi}(t, t_0) \, f^0(a - (t - t_0)), & a \geq t - t_0 \end{cases}$$

where $\mathbf{\Phi}$ is a state-transition matrix, and $f^0$ is an initial age density distribution at initial time $t_0$. We obtain $\mathbf{\Phi}$ by taking advantage of an existing numerical solution $x(t)$, which we plug in the original system, obtaining a new compartmental matrix $\tilde{\mathbf{B}}(t) := \mathbf{B}(x(t), t)$ and a new input vector $\tilde{u} := u(x(t), t)$. Then, the new linear non-autonomous compartmental system

$$\dot{y}(t) = \tilde{\mathbf{B}}(t) \, y(t) + \tilde{u}(t), \quad t > t_0, \tag{B6}$$

has the unique solution $y(t) = x(t)$, which emerges from the fact that both systems are identical. The solution of the system is then given by

$$x(t) = \mathbf{\Phi}(t, t_0) \, x^0 + \int_{t_0}^{t} \mathbf{\Phi}(t, s) \, u(s) \, \mathrm{d}s, \tag{B7}$$





where $x^0 = \int_0^\infty f^0(a)\,\mathrm{d}a$ is the initial vector of carbon stocks. We obtain the state-transition matrix as the solution of the following matrix differential equation

$$\frac{\mathbf{\Phi}(t,t_0)}{\mathrm{d}t} = \mathbf{B}(t)\,\mathbf{\Phi}(t,t_0), \quad t > t_0, \tag{B8}$$

with initial condition

$$\mathbf{\Phi}(t_0,t_0) = \mathbf{I}, \tag{B9}$$

where $\mathbf{I} \in \mathbb{R}^{n \times n}$ is the identity matrix.

These formulas can be applied to any carbon cycle model represented as a compartmental system to obtain the fate of carbon once it enters the ecosystem as well as timescale metrics such as age and transit time distributions.

**Computation of the mass remaining in the system**

From equation (B7), we can see from the first term that the initial amount of carbon in the system $x^0$ changes over time according to the term $\mathbf{\Phi}(t,t_0)\,x^0$. Rasmussen et al. (2016) showed that under certain circumstances, equation (B7) is exponentially stable as long as $\mathbf{B}$ is invertible, and the state transition operator acts as a term that exponentially 'decomposes' the initial amount of carbon. Furthermore, the state transition operator tracks the dynamics of the incoming carbon and how it is transferred among the different pools before it is respired. Therefore, this operator can be used to compute the fate of an amount of carbon sequestered at time $t_s$ as

$$M_s(t - t_s) = M_s(a) = \|\mathbf{\Phi}(t,t_s)\,u(t_s)\|, \quad a = t - t_s. \tag{B10}$$

Similarly, the fate of one unit of sequestered carbon at time $t_s$ can be computed as

$$M_{s1}(a) = \left\| \mathbf{\Phi}(t,t_s) \cdot \frac{u(t_s)}{\|u(t_s)\|} \right\|, \tag{B11}$$

where the subscript 1 denotes that the function predicts the fate of one unit of carbon.





## Appendix C: Detailed representation of the modified models used in examples

For the scenario S1 with high efficiency in transfers to the long-duration products, the matrix of cycling rates is given by

$$
\mathbf{B} = \begin{pmatrix}
-2.38 & 0.00 & 0.00 & 0.00 & 0.00 & 0.00 & 0.00 & 0.00 & 0.00 & 0.00 \\
0.84 & -0.37 & 0.00 & 0.00 & 0.00 & 0.00 & 0.00 & 0.00 & 0.00 & 0.00 \\
0.00 & 0.00 & -0.52 & 0.00 & 0.00 & 0.00 & 0.00 & 0.00 & 0.00 & 0.00 \\
0.86 & 0.04 & 0.17 & -0.59 & 0.00 & 0.00 & 0.00 & 0.00 & 0.00 & 0.00 \\
0.00 & 0.00 & 0.09 & 0.04 & -0.01 & 0.00 & 0.00 & 0.00 & 0.00 & 0.00 \\
0.00 & 0.18 & 0.00 & 0.00 & 0.00 & -0.02 & 0.00 & 0.00 & 0.00 & 0.00 \\
0.00 & 0.06 & 0.00 & 0.00 & 0.00 & 0.00 & -0.04 & 0.00 & 0.00 & 0.00 \\
0.00 & 0.03 & 0.00 & 0.00 & 0.00 & 0.00 & 0.00 & -0.32 & 0.00 & 0.00 \\
0.00 & 0.01 & 0.00 & 0.00 & 0.00 & 0.00 & 0.00 & 0.00 & -0.70 & 0.00 \\
0.00 & 0.01 & 0.00 & 0.00 & 0.00 & 0.00 & 0.00 & 0.00 & 0.00 & -1.00
\end{pmatrix}, \tag{C1}
$$

which results in a steady-state vector of pool sizes given by

$$
x^* = \begin{pmatrix}
32.38 \\
72.72 \\
69.00 \\
72.12 \\
876.59 \\
654.48 \\
111.88 \\
6.82 \\
1.56 \\
1.09
\end{pmatrix}, \quad \|x^*\| = 1898.62. \tag{C2}
$$





For the scenario S2 of low efficiency transfers to long-duration pools, the cycling rate matrix is given by

$$
\mathbf{B} = \begin{pmatrix}
-2.38 & 0.00 & 0.00 & 0.00 & 0.00 & 0.00 & 0.00 & 0.00 & 0.00 & 0.00 \\
0.84 & -0.37 & 0.00 & 0.00 & 0.00 & 0.00 & 0.00 & 0.00 & 0.00 & 0.00 \\
0.00 & 0.00 & -0.52 & 0.00 & 0.00 & 0.00 & 0.00 & 0.00 & 0.00 & 0.00 \\
0.86 & 0.04 & 0.17 & -0.59 & 0.00 & 0.00 & 0.00 & 0.00 & 0.00 & 0.00 \\
0.00 & 0.00 & 0.09 & 0.04 & -0.01 & 0.00 & 0.00 & 0.00 & 0.00 & 0.00 \\
0.00 & 0.09 & 0.00 & 0.00 & 0.00 & -0.02 & 0.00 & 0.00 & 0.00 & 0.00 \\
0.00 & 0.06 & 0.00 & 0.00 & 0.00 & 0.00 & -0.04 & 0.00 & 0.00 & 0.00 \\
0.00 & 0.03 & 0.00 & 0.00 & 0.00 & 0.00 & 0.00 & -0.32 & 0.00 & 0.00 \\
0.00 & 0.09 & 0.00 & 0.00 & 0.00 & 0.00 & 0.00 & 0.00 & -0.70 & 0.00 \\
0.00 & 0.03 & 0.00 & 0.00 & 0.00 & 0.00 & 0.00 & 0.00 & 0.00 & -1.00
\end{pmatrix}, \tag{C3}
$$

which results in a steady-state vector of pools sizes as

$$
x^* = \begin{pmatrix}
32.38 \\
72.72 \\
69.00 \\
72.12 \\
876.59 \\
327.24 \\
111.88 \\
6.82 \\
9.35 \\
2.18
\end{pmatrix}, \quad \|x^*\| = 1580.26. \tag{C4}
$$

For scenario S3 of low efficiency transfers and improved silviculture, the matrix of cycling rates is given by

$$
\mathbf{B} = \begin{pmatrix}
-2.08 & 0.00 & 0.00 & 0.00 & 0.00 & 0.00 & 0.00 & 0.00 & 0.00 & 0.00 \\
1.11 & -0.37 & 0.00 & 0.00 & 0.00 & 0.00 & 0.00 & 0.00 & 0.00 & 0.00 \\
0.00 & 0.00 & -0.52 & 0.00 & 0.00 & 0.00 & 0.00 & 0.00 & 0.00 & 0.00 \\
0.30 & 0.04 & 0.17 & -0.59 & 0.00 & 0.00 & 0.00 & 0.00 & 0.00 & 0.00 \\
0.00 & 0.00 & 0.09 & 0.04 & -0.01 & 0.00 & 0.00 & 0.00 & 0.00 & 0.00 \\
0.00 & 0.09 & 0.00 & 0.00 & 0.00 & -0.02 & 0.00 & 0.00 & 0.00 & 0.00 \\
0.00 & 0.06 & 0.00 & 0.00 & 0.00 & 0.00 & -0.04 & 0.00 & 0.00 & 0.00 \\
0.00 & 0.03 & 0.00 & 0.00 & 0.00 & 0.00 & 0.00 & -0.32 & 0.00 & 0.00 \\
0.00 & 0.09 & 0.00 & 0.00 & 0.00 & 0.00 & 0.00 & 0.00 & -0.70 & 0.00 \\
0.00 & 0.03 & 0.00 & 0.00 & 0.00 & 0.00 & 0.00 & 0.00 & 0.00 & -1.00
\end{pmatrix}, \tag{C5}
$$



which leads to the vector of steady-state contents

$$x^* = \begin{pmatrix} 37.00 \\ 109.92 \\ 69.00 \\ 45.79 \\ 797.59 \\ 494.63 \\ 169.10 \\ 10.30 \\ 14.13 \\ 3.30 \end{pmatrix}, \quad \|x^*\| = 1750.76. \tag{C6}$$

The modified version of the model in which the active soil pool is replaced by the pool structure of the Century model has the following matrix of cycling rates

$$\mathbf{B} = \begin{pmatrix} -2.08108 & 0.00000 & 0.00000 & 0.00000 & 0.00000 & 0.00000 & 0.00000 \\ 0.83784 & -0.06858 & 0.00000 & 0.00000 & 0.00000 & 0.00000 & 0.00000 \\ 0.00000 & 0.00000 & -0.52174 & 0.00000 & 0.00000 & 0.00000 & 0.00000 \\ 0.56757 & 0.03319 & 0.17391 & -0.59259 & 0.00000 & 0.00000 & 0.00000 \\ 0.00000 & 0.00442 & 0.08696 & 0.03704 & -0.07175 & 0.00160 & 0.00006 \\ 0.00000 & 0.00000 & 0.00000 & 0.00000 & 0.04219 & -0.00380 & 0.00000 \\ 0.00000 & 0.00000 & 0.00000 & 0.00000 & 0.00029 & 0.00011 & -0.00013 \end{pmatrix}, \tag{C7}$$

which results in a steady-state vector of pools sizes as

$$x^* = \begin{pmatrix} 37.00 \\ 452.00 \\ 69.00 \\ 81.00 \\ 206.26 \\ 2289.92 \\ 2463.44 \end{pmatrix}, \quad \|x^*\| = 5598.616. \tag{C8}$$

*Author contributions.* CAS conceived the idea and wrote the manuscript, SEC and EDS contributed the examples, MH and HM helped to develop the mathematical framework. All authors contributed to writing.

*Competing interests.* No competing interests to declare





*Acknowledgements.* Funding was provided by the Max Planck Society and the German Research Foundation (Emmy-Noether Programme SI 1953/2-2). We would like to thank Mark Harmon, Yiqi Luo, and Susan Trumbore for useful comments on previous versions of this

manuscript.



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

$$AGWP = \int\limits_{t_0}^{t_0+T} k_{CO_2} \cdot M_a(t)\, dt$$

CO₂

$M_a(t)$

Time since emission

$$CBS = \int\limits_{t_0}^{t_0+T} k_{CO_2} \cdot M_a'(t)\, dt$$

CO₂

$M_a'(t)$

Time since sequestration

CO₂ emission

*Atmosphere*

CO₂ release

CO₂ uptake

*Land*

CO₂ uptake

Fate of ecosystem C
not considered

$$CS = \int\limits_{t_0}^{t_0+T} M_s(t)\, dt$$

Ecosystem
C

$M_s(t)$

Time since sequestration

**Figure 1.** Contrast between current approach to quantification of climate effects of emissions and sequestration (left), and the proposed approach for sequestration (right). Plots and equations represent the concepts of absolute global warming potential (AGWP) of an emission of $CO_2$, carbon sequestration (CS), and climate benefits of sequestration (CBS). AGWP integrates over a time horizon $T$ the fate of an instant emission at time $t_0$ of a gas ($M_a(t)$) and multiplies by the radiative efficiency $k$ of the gas. A similar idea can be used to define CS as the integral of the fate $M_s(t)$ of an instant amount of carbon uptake $S_0$ over $T$. The CBS captures the atmospheric 'disturbance' caused by $CO_2$ uptake and subsequent release by respiration as the integral over $T$ of the fate of sequestered carbon $M_a'(t)$ multiplied by the radiative efficiency of $CO_2$.





**Figure 2.** Impulse response function of carbon dioxide in the atmosphere $(h_a(t))$ as predicted by different authors. Joos PD100 represents the impulse response experiment reported in Joos et al. (2013) for a present day (PD) atmosphere with a 100 PgC emission pulse. Joos PI100 represents an impulse response experiment from the same study using a pulse of 100 PgC in a pre-industrial (PI) atmosphere. In addition, we include the impulse response curve reported in Lashof and Ahuja (1990).



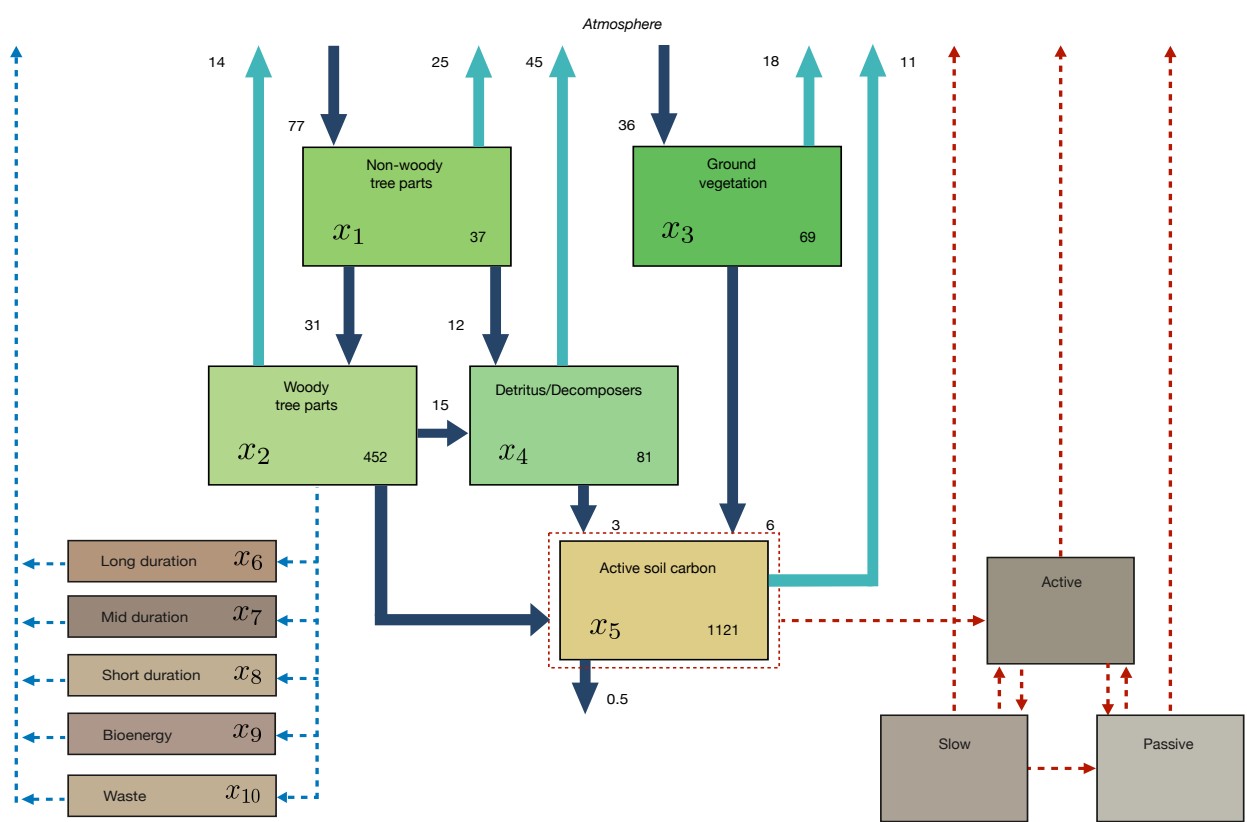

**Figure 3.** Graphical representation of the terrestrial carbon cycle model developed by Emanuel et al. (1981) with the modified structure used for the examples presented in this manuscript. The original model is presented in the center of the figure in black with respective values for stocks and fluxes. The modified version used in Example 2 introduced the wood-product pools presented in blue on the left of the diagram. The modified version used in Example 3, replaces the active soil pool by the pool structure used in the Century model.



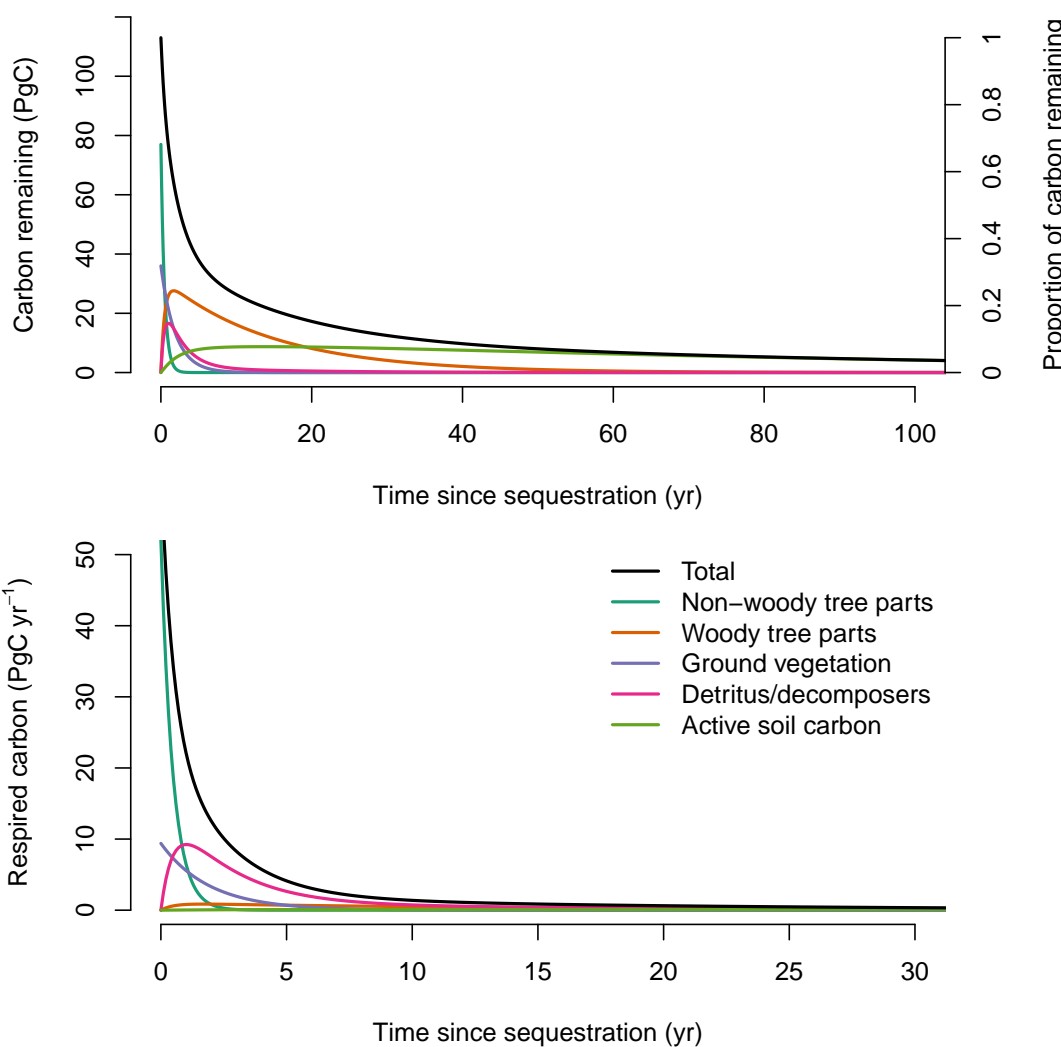

**Figure 4.** Fate of carbon ($M_s(t)$ left axis and $M_{s1}(t)$ right axis) entering the terrestrial biosphere according to the model of Emanuel et al. (1981) calculated using equation (13) for the upper panel, and respired carbon ($r(t)$) returning back to the atmosphere.

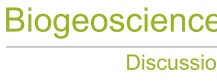
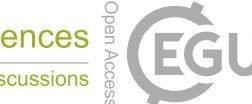

**Figure 5.** Carbon sequestration (CS), climate benefit of sequestration (CBS), and absolute global warming potential (AGWP) for instantaneous uptake or emissions. a) CS due to the uptake of 113 PgC, b) CS due to the uptake of one unit of carbon ($CS_1$). c) CBS due to the uptake of 113 PgC for two different impulse response functions (IRF). d) AGWP due to the emission of 1 $PgCO_2$-C to the atmosphere for two different IRFs, the pre-industrial experiment from Joos et al. (2013) and the function reported in Lashof and Ahuja (1990). Dashed lines in panels a and b represent the steady-state carbon storage and mean transit time, respectively.



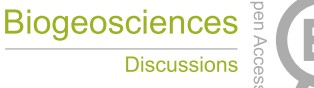

**Figure 6.** Different carbon management strategies and their effect on the CS and $CS_1$. Management to increase or decrease carbon inputs in the vector $u$ by specific proportions $\gamma$ are shown in panels a and b. Management to increase or decrease process rates in the matrix $\mathbf{B}$ by a proportion $\xi$ are shown in panels c and d. Since $CS_1$ quantifies carbon sequestration of one unit of carbon, management of the amount of carbon inputs does not modify $CS_1$ in panel b, and all lines overlap.



**Figure 7.** Effects of different management strategies on CBS. The upper panel shows the effect of increasing or decreasing carbon inputs by a proportion $\gamma$, while the lower panel shows the effects of decreasing or increasing process rates in the matrix **B** by a proportion $\xi$.







**Figure 8.** Simulations of time-dependent CS and CBS for the industrial period using the model of Emanuel et al. (1981) driven by the average GPP of models from the CMIP6 archive (*esm-hist* experiment). a) Average GPP and predicted respiration fluxes of the input and their difference. b) Amount of carbon remaining of the inputs for every year calculated using equation (25). The areas under the curve are values of CS for time horizons of 50 yr starting in 1850, 1900, and 1950, computed using equation (24). c) Continuous values of CS (areas under the curve of panel b for time horizons from 1 to 150 yr. The dash line represents CS for the equilibrium case with constant inputs of 113 PgC yr$^{-1}$. d) Continuous values of CBS for time horizons from 1 to 150 yr computed using equation (28).





**Figure 9.** Comparison of the fate of carbon $M_s(t)$, carbon sequestration CS, and the climate benefit of carbon sequestration CBS of an instantaneous uptake of 113 PgC among different management scenarios for forest products. Plots on the left (a, c, d) show the values obtained for the no management case and the three scenarios, while panels on the right (b, d, f) show the difference between each scenario and the no-management case. S1 has a high efficiency of transfer to long-duration products and S2 a low efficiency of transfers. S3 includes low efficiencies in transfers as in S2, but increases allocation to wood due to silvicultural management. Colors in all plots follow the same legend as in a.



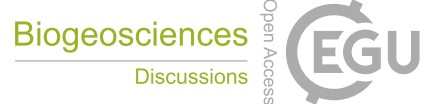



**Figure 10.** Fate of carbon in the bioenergy pool for the three scenarios (a), and the corresponding climate benefit of sequestration CBS for this pool (b).







**Figure 11.** Effect of replacing the active soil pool of the Emanuel et al. (1981) model with the pool structure of the Century model described in Parton et al. (1987). a) Differences in the proportion of carbon remaining in the model with soil structure according to Century versus the original model, b) carbon sequestration for all scenarios, and c) climate benefit of sequestration for all scenarios.