# Peer review of "The Climate Benefit of Carbon Sequestration"

_Biogeosciences, 2020_

## Referee Comment (RC1) · Alexey Shiklomanov (Referee) · 13 Jul 2020

Sierra et al. define a new metric—the climate benefit of sequestration (CBS)—for analyzing the carbon-climate system. This metric is roughly analogous to global warming potential (GWP) but for carbon sequestered in the land surface (e.g. vegetation, soil); similarly to GWP, it reflects both the amount and typical lifetime of carbon in these surface pools, with higher CBS for systems that sequester more carbon and/or hold onto that carbon for longer. Sierra et al. then discuss the usage and implications of this metric in several different cases: A simple matrix-based global carbon cycle model of the pre-industrial world; implications of CBS for management; investigation of globally-averaged CBS during the industrial period (1850-present); and global applications of CBS with respect to forests and soils.

The underlying idea of CBS makes sense, and the authors are to be commended for

describing CBS and how they applied it to their various cases in detail. However, I do not think this manuscript makes a meaningful contribution to the literature on climate or carbon cycling. To justify the development of this metric, the authors would need to make a compelling case for where existing, widely-used related metrics of carbon sequestration (such as carbon residence time and net ecosystem productivity) fall short, and why this new metric is superior (or at least complementary). However, there is hardly any mention of these existing metrics in this manuscript. For that matter, the manuscript's application of CBS to terrestrial C cycle models focuses only on dated and/or simplistic models and makes no mention of recent syntheses of terrestrial carbon cycling and associated climate feedbacks by modern land surface models—for instance, Friedlingstein et al. 2014 (https://doi.org/10.1175/JCLI-D-12-00579.1) or Heinze et al. 2019 (https://doi.org/10.5194/esd-10-379-2019)—which makes it difficult to draw meaningful conclusions from those results.

Ultimately, I cannot recommend this manuscript for publication beyond the Discussion format. I would encourage the authors to carefully read a recent review of terrestrial C cycling and its climate implications and consider how CBS fits into that context.

---

## Referee Comment (RC2) · Anonymous Referee #2 · 18 Jul 2020

Overall: I think this is an interesting paper, and lays a simple and elegant formalism for understanding the importance of changes to ecosystem carbon flows from the perspective of atmospheric radiative balance. However, the applications of the method are more confusing than helpful, and don't really sell the utility of the method for answering policy-relevant questions. So my overall suggestions here are major revisions along the lines of: (1) take some time in section 2 to explain a bit more what each of the terms here mean, (2) reformulate the subsequent examples to use more real-world numbers that describe specific sequestration activities: afforestation, deforestation, changed agricultural practices, etc, where the comparison is made between a perturbed and unperturbed ecosystem; (3) concentrate less on the long-term dynamics and more on the comparison of unperturbed vs perturbed ecosystem changes over discrete and policy-relevant time horizons.

Section 2.3. I think more detail needs to be given here for how this method works when

the linear and/or equilibrium assumptions are not justified.

Sections 3-4. I think some detail is needed on what exactly the proposed sequestration that is being modeled here is. It seems like the sequestration proposed here is to create a new ecosystem where none existed previously, so that x(t=0) = 0 and u is being changed from zero to some global-mean vlaue. But a typical sequestration plan is to afforest a given patch of ground, i.e. converting from grassland or crops to forest. How would such a transition, where the change is to both the u vector and B matrix when x(t=0) != 0, be calculated?

Line 245: I disagree with the authors here and think that if they are going to go this route, then they need to justify their decision much more than they do. Of course carbon does return to geologic reservoirs, but the timescale of this is much longer than the 10-1000 year timescale discussed here. So I think, if anything, the Joos et al curves should be used and the Lashof & Aruha ones removed. I'm honestly confused about why the authors would suggest the opposite.

Line 265: I disagree with the idea that changing u will, in general, not lead to a change in B. I think there is quite a bit of evidence (forest self-thinning, soil carbon saturation) that B is highly sensitive to changes in u in real ecosystems This comes up again a few paragraphs later. While it is mathematically convenient to separate these two things, I think in general it is not really possible to change without the other (nor a priori to assert what sign that change to u or B will necessarily be).

Line 293: This is a fairly obvious result and so I'm not sure why this formalism is needed to make that point?

Section 5. I am not sure I understand the point of this example, and I also think there is a conceptual error being made here when the method is applied to large (relative to the total biospheric fluxes) sequestration perturbations: as I understand the notation used here, the function h_a(t) represents the remaining pulse (positive or negative) of CO2 into the atmosphere. But much (∼50%) of the loss of that atmospheric concentration

pulse is due to the beta effect of land carbon responding to the carbon that has been emitted. So it seems like you are double counting this biospheric response, as it appears in both h_a(t) and in r(t)? I suspect this whole approach only works for small perturbations to the biosphere, where h_a(t) and r(t) are approximately non-overlapping, and thus excludes this example here.

Table 1. Where are these numbers coming from? It seems like the authors are just sort of making them up as heuristic examples. Is that the case, and if so, might it be more useful to use numbers based on real-world, even if highly simplified, examples? Similarly, the u and B numbers from Emanuel et al (1980) are for a globally-averaged ecosystem? If so, I think this wouldn't make sense for this example and you would have to use examples for a specific forest ecosystem instead. I understand the intention here is to be heuristic but more realistic numbers shouldn't be too hard to track down and it'd be infomrative to try to do something that corresponds more to reality when talking about concrete examples such as this.

Line 395. Can they? I see how albedo could, but other surface energy terms or surface roughness imply a tradeoff of one or another type of energy, or redistributions of energy between the land and atmosphere, and so can't really be compared to this metric.

Section 7: I'm not totally sure how comparing the CBS metric of two extremely vintage ecosystem models (one of which is a global-mean number and the other is a sort of reference-temperature number, so not really comparable even) is really of any importance to the argument being made in this manuscript, or anything else really. If the point is just that soils store a lot of carbon for a long time, don't we already know that? Suggest substantially revising or deleting this section.

Line 476-484. The other (much larger, really) problem with GWP is that its utility completely depends on what time interval the metric is integrated over; hence the unending debates about how much policy should focus on CH4 as compared to CO2. This problem applies equally to the CBS, but is completely skipped over in this paper. How

would the CBS be used in a policy-relevant context where we care about limiting peak temperature at some time period? Are there sequestration methods that are positive at a 50-year time horizon but negative on shorter or longer timescales? Is there a CBS analogue for GWP-star? Exploring these question would seem to be central to how this metric would actually be used in practice, but isn't actually touched on at all in this manuscript. I think this is a mistake and a major shortcoming of the current manuscript.

Lines 501-509. These are really not trivial problems, and substantially degrade the utility of this metric. The criticism of the Joos et al model strikes me as wildly off base; the irreversibility of global warming on shorter than multi-millenial timescales is a core feature of the problem and so asserting it away as something that can be ignored is not a good idea. In principle, the uncertainty in the impulse response function would be the same if used for two separate treatments, i.e. a baseline and a perturbed ecosystem, thus it seems like the more useful application of this method would be as an analog to GWP (not AGWP): calculate CBS of both a directly-perturbed ecosystem and an unperturbed ecosystem (or relatively unperturbed, i.e. not logged or afforested or whatever the treatment is, but still subject to CO2 fertilization, changes in climate-driven mortality, etc), recognizing that, in a globally changed world, neither will likely be at steady state, and calculate a relative CBS as the ratio of the two absolute CBS.

---

## Author Comment (AC1) · 7 Aug 2020

We thank the reviewer for taking the time to review and comment on our manuscript, and regret that he finds this manuscript 'not a meaningful contribution to the climate literature'. We understand that in the current version of the manuscript, the link to state-of-the-art Earth system models (ESMs) is not completely clear. However, we would like to clarify that the concepts we introduce are very general and can deal with models of any level of complexity, working at different scales, and developed for different purposes. The compartmental approach we use works for simple 'outdated' models as well as for highly complex models. We have written extensively in the past about this generalization approach (Sierra and Müller, 2015; Metzler et al., 2018; Sierra et al., 2018; Ceballos-Núñez et al., 2020), and similar ideas are also well developed in papers by Dr. Yiqi Luo's group (e.g Luo and Weng, 2011; Xia et al., 2013; Luo et al., 2017). Unfortunately, we do not have much space in this manuscript to demonstrate the

generality of the compartmental framework, but we have done exactly that in previous publications. Similarly for the limitations of the concept 'residence time', for which the reviewer asks why we do not show the limitations of such a concept. We also have written a number of publications about issues with this concept and the common methods to compute it (ratio of stock over flux). We developed new mathematical approaches to advance on this subject (Metzler and Sierra, 2018; Metzler et al., 2018) and this manuscript is a step further in the application of the new methods. Again, we feel this manuscript is not the appropriate venue to elaborate on the 'residence time' issues and the new methods. Instead, we provide a presentation on these ideas, show with more detail some of the formulas in the appendix, and provide appropriate references, but can't go in more detail. Also, we want to point out that the problem on how to account for time in carbon sequestration has been a long-standing issue, with important debates in forestry (e.g. Fearnside, 1995; Fearnside et al., 2000; Sedjo and Sohngen, 2012), ecological economics (e.g. Moura Costa and Wilson, 2000), and ecosystem management (e.g. Neubauer and Megonigal, 2015). We think we provide here a relevant contribution to those previous debates. This may not be very obvious for researchers currently working on climate feedbacks, but it is a topic that touches on different disciplines and we think it is a meaningful contribution to the overall topic of carbon sequestration in natural and anthropogenic sinks.

Below, we provide specific answers to the main issues raised by the reviewer (in *italics*), with a description of corresponding changes in the manuscript to address those comments.

Answers to specific comments

- *To justify the development of this metric, the authors would need to make a compelling case for where existing, widely-used related metrics of carbon sequestration (such as carbon residence time and net ecosystem productivity) fall short,*

*and why this new metric is superior (or at least complementary). However, there is hardly any mention of these existing metrics in this manuscript.*

We discuss in our manuscript the limitations of the use of global warming potential (GWP), which is the most common metric to assess climate consequences of carbon management. Limitations of the concept of residence times have been already published in Sierra et al. (2017), where we elaborate on the need to distinguish between the concepts of system age and transit time. The present manuscript is a further development of the concept of transit time to show that it can be used to quantify the climate benefit of having carbon stored in ecosystems during the time it remains there. Therefore, we focus not in showing the limitations of the ambiguous concept of residence time, but rather in showing the power of the transit time/age framework. The manuscript did not elaborate on limitations of the concept of net ecosystem production (NEP) to assess carbon sequestration, although its limitations are somehow intuitive based on Fig 1 and the text in the introduction. NEP provides a net flux between the carbon exchanged between ecosystems and the atmosphere, without accounting for harvest exports. It does not tell you for how long the carbon in the output flux stayed in the ecosystem. Two different ecosystems with similar NEP values could have very different carbon storage values and transit times, so this concept is not very useful to assess carbon sequestration, particularly for long time scales.

To address these issues, we added text in the introduction briefly mentioning our previous work on the ambiguity and limitations of 'residence time'. We also elaborate more on the limitations of studying either gross (e.g. GPP) or net (NEP) fluxes to assess carbon sequestration.

We also would like to mention that our CBS concept just tells something different that other metrics do not tell. It combines in a single metric the amount of carbon that enters a sink and the time it remains there. Previous metrics simply do not provide this information in an integrated form.

- *For that matter, the manuscript's application of CBS to terrestrial C cycle models focuses only on dated and/or simplistic models and makes no mention of recent syntheses of terrestrial carbon cycling and associated climate feedbacks by modern land surface models—for instance, Friedlingstein et al. 2014 (https://doi.org/10.1175/JCLI-D-12-00579.1) or Heinze et al. 2019 (https://doi.org/10.5194/esd-10-379-2019)—which makes it difficult to draw meaningful conclusions from those results.*

We chose a 'dated' model because, to introduce a new complex metric, we believe a simple model is more effective and transparent than a complex Earth system model. With an ESM, potential users of this framework would not have the opportunity to test results if they do not have access to the source code of the ESM and to a supercomputer. A simple model allows readers to test the framework with very simple code. We believe this is a more transparent approach for a paper that introduces a new concept. Future applications can of course be implemented in large ESMs, but that is beyond the scope of this manuscript.

However, a simple model like the one we used does not have any feedbacks with other components of the Earth system. Nevertheless, these feedbacks can be part of the atmospheric impulse response function. In fact, the IRFs of Joos et al. (2013) include all feedbacks that are part of the ESMs of the CMIP5 generation. If one uses these IRFs for the computation of the CBS, it is not necessary to include feedbacks in the terrestrial carbon model as long as simulations do not deviate much from the original simulations used to create the IRF. Alternatively, one can use an ESM and compute the CBS directly from the net biospheric fluxes and the particular IRF of the model. This would require a complex simulation setup, that again is beyond the scope of this manuscript. More importantly, we would like to emphasize that our aim is not to produce state-of-the-art calculations of the CBS for the terrestrial biosphere, but rather to introduce the concept and the mathematical theory behind it. For this reason, and also based on the comments

from reviewer 2, we decided to remove some of the examples and give more emphasis to the development and explanation of the concepts. After all, this is a theoretical paper and not a state-of-the-art analysis of the actual climate benefit of sequestering carbon in the terrestrial biosphere.

- *Ultimately, I cannot recommend this manuscript for publication beyond the Discussion format. I would encourage the authors to carefully read a recent review of terrestrial C cycling and its climate implications and consider how CBS fits into that context*

We will check the recent carbon cycle literature to see if we are missing something. We have coauthored a good number of papers on the carbon cycle, including reviews. But indeed there might be something we are missing. We would have hoped the reviewer to point out more clearly what exactly are we missing. Without a clear indication of what particular concepts are problematic in our framework, it is impossible for us to guess them.

**References**

Ceballos-Núñez, V., Müller, M., and Sierra, C. A. (2020). Towards better representations of carbon allocation in vegetation: a conceptual framework and mathematical tool. *Theoretical Ecology*, in press.

Fearnside, P. M. (1995). Global warming response options in Brazil's forest sector: Comparison of project-level costs and benefits. *Biomass and Bioenergy*, 8(5):309 – 322. Forestry and Climate Change.

Fearnside, P. M., Lashof, D. A., and Moura-Costa, P. (2000). Accounting for time in mitigating global warming through land-use change and forestry. *Mitigation and Adaptation Strategies for Global Change*, 5(3):239–270.

Joos, F., Roth, R., Fuglestvedt, J. S., Peters, G. P., Enting, I. G., von Bloh, W., Brovkin, V., Burke, E. J., Eby, M., Edwards, N. R., Friedrich, T., Frölicher, T. L., Halloran, P. R., Holden, P. B., Jones, C., Kleinen, T., Mackenzie, F. T., Matsumoto, K., Meinshausen, M., Plattner,

G.-K., Reisinger, A., Segschneider, J., Shaffer, G., Steinacher, M., Strassmann, K., Tanaka, K., Timmermann, A., and Weaver, A. J. (2013). Carbon dioxide and climate impulse response functions for the computation of greenhouse gas metrics: a multi-model analysis. *Atmospheric Chemistry and Physics*, 13(5):2793–2825.

Luo, Y., Shi, Z., Lu, X., Xia, J., Liang, J., Jiang, J., Wang, Y., Smith, M. J., Jiang, L., Ahlström, A., Chen, B., Hararuk, O., Hastings, A., Hoffman, F., Medlyn, B., Niu, S., Rasmussen, M., Todd-Brown, K., and Wang, Y.-P. (2017). Transient dynamics of terrestrial carbon storage: mathematical foundation and its applications. *Biogeosciences*, 14(1):145–161.

Luo, Y. and Weng, E. (2011). Dynamic disequilibrium of the terrestrial carbon cycle under global change. *Trends in Ecology & Evolution*, 26(2):96–104.

Metzler, H., Müller, M., and Sierra, C. A. (2018). Transit-time and age distributions for nonlinear time-dependent compartmental systems. *Proceedings of the National Academy of Sciences*, 115(6):1150–1155.

Metzler, H. and Sierra, C. A. (2018). Linear autonomous compartmental models as continuous-time Markov chains: Transit-time and age distributions. *Mathematical Geosciences*, 50(1):1–34.

Moura Costa, P. and Wilson, C. (2000). An equivalence factor between $CO_2$ avoided emissions and sequestration – description and applications in forestry. *Mitigation and Adaptation Strategies for Global Change*, 5(1):51–60.

Neubauer, S. C. and Megonigal, J. P. (2015). Moving beyond global warming potentials to quantify the climatic role of ecosystems. *Ecosystems*, 18(6):1000–1013.

Sedjo, R. and Sohngen, B. (2012). Carbon sequestration in forests and soils. *Annual Review of Resource Economics*, 4(1):127–144.

Sierra, C. A. and Müller, M. (2015). A general mathematical framework for representing soil organic matter dynamics. *Ecological Monographs*, 85:505–524.

Sierra, C. A., Müller, M., Metzler, H., Manzoni, S., and Trumbore, S. E. (2017). The muddle of ages, turnover, transit, and residence times in the carbon cycle. *Global Change Biology*, 23(5):1763–1773.

Sierra, C. A., Ceballos-Núñez, V., Metzler, H., and Müller, M. (2018). Representing and understanding the carbon cycle using the theory of compartmental dynamical systems. *Journal of Advances in Modeling Earth Systems*, 10(8):1729–1734.

Xia, J., Luo, Y., Wang, Y.-P., and Hararuk, O. (2013). Traceable components of terrestrial carbon storage capacity in biogeochemical models. *Global Change Biology*, 19(7):2104–2116.

---

## Author Comment (AC2) · 7 Aug 2020

We appreciate the thoughtful comments from the Reviewer and the critical questions she/he poses. Based on this review, we decided to make a major restructuring of the manuscript. First, we decided to give more emphasis to the theoretical part of the manuscript and decrease the emphasis on the importance of the examples. Therefore, we show now very simple examples on the computation of the CBS, still using a simple model for clarity and transparency, and discussing potential applications in the Discussion section. Second, we use now a different model, slightly updated, focused on the ecosystem level and not on the biospheric level. It is still a very simple model, which has the advantage that it is tractable and the computations can be reproduced in a transparent way.

Below, we elaborate more on the proposed changes, and provide specific answers to

the reviewer's comments (in *italics*).

**Answers to general comments**

- *Overall: I think this is an interesting paper, and lays a simple and elegant formalism for understanding the importance of changes to ecosystem carbon flows from the perspective of atmospheric radiative balance. However, the applications of the method are more confusing than helpful, and don't really sell the utility of the method for answering policy-relevant questions.*

  This is an important comment and we took it seriously. We acknowledge that the examples we provided are not necessarily very useful in explaining the main CBS concept and showing its relevance for policy-related questions. In the new revised version, the examples were completely changed, they now illustrate how these quantities are computed and what are the meaning of the obtained results. Direct applications to policy are addressed in the Discussion section. We realized that our original examples can be topics of importance to be addressed in separate studies, but the use of our simple model creates an imbalance of addressing an important topic with a very simple model. Therefore, we think those particular questions can be addressed in subsequent manuscripts that use more appropriate models for the questions being asked.

- *So my overall suggestions here are major revisions along the lines of: (1) take some time in section 2 to explain a bit more what each of the terms here mean, (2) reformulate the subsequent examples to use more real-world numbers that describe specific sequestration activities: afforestation, deforestation, changed agricultural practices, etc, where the comparison is made between a perturbed and unperturbed ecosystem; (3) concentrate less on the long-term dynamics and more on the comparison of unperturbed vs perturbed ecosystem changes over*

*discrete and policy-relevant time horizons.*

We made major revisions following these recommendations. In particular, 1) we expanded section 2 to better explain the terms of the equations and their potential use. Also, we made an effort to better explain the special cases of the framework and how they can be used for particular problems. 2) We changed considerably the examples to avoid misunderstandings in that we do not provide definite answers to particular questions. Rather, we use examples that show how the framework is used for particular *computations* that could be helpful for some particular problems addressed in the Discussion section.

Answers to specific comments

- *Section 2.3. I think more detail needs to be given here for how this method works when the linear and/or equilibrium assumptions are not justified*

  More details are giving here and also in section 2.4 on how to apply the concepts to the non-equilibrium case.

- *Sections 3-4. I think some detail is needed on what exactly the proposed sequestration that is being modeled here is. It seems like the sequestration proposed here is to create a new ecosystem where none existed previously, so that x(t=0) = 0 and u is being changed from zero to some global-mean value. But a typical sequestration plan is to afforest a given patch of ground, i.e. converting from grassland or crops to forest. How would such a transition, where the change is to both the u vector and B matrix when x(t=0) != 0, be calculated?*

  The analyses presented in sections 3 and 4 are steady-state analyses and not an afforestation case. The idea here is to show what is the climate benefit of an amount of carbon taken up by an ecosystem that is already in equilibrium. For example, it can be used to calculate carbon sequestration as an ecosystem

service, say an old-growth tropical forest and how much warming is avoided over the long-term. We give now more details about this and explain cases in which these analyses might be justified.

- *Line 245: I disagree with the authors here and think that if they are going to go this route, then they need to justify their decision much more than they do. Of course carbon does return to geologic reservoirs, but the timescale of this is much longer than the 10-1000 year timescale discussed here. So I think, if anything, the Joos et al curves should be used and the Lashof & Aruha ones removed. I'm honestly confused about why the authors would suggest the opposite.*

We agree with the reviewer and we are aware that this manuscript is not the best place to challenge the IRFs of Joos et al. (2013). However, it is still problematic to use this IRF for long-term and steady-state analyses. The decision of Joos et al. (2013) to introduce an intercept term with an infinity timescale in their IRFs is not well addressed in their manuscript. This is something not relevant for many analyses focused on policy-relevant timescales, but it is relevant for steady-state analyses as we showed in the previous version of the manuscript. More recently, Millar et al. (2017) addressed this issue by simply assigning a timescale of 1 million years to the proportion of carbon that in the IRFs of Joos et al. (2013) had an infinity timescale. We decided to follow the same approach as Millar et al. (2017), still using the same timescales of Joos et al. (2013), but avoiding entering in a discussion of appropriate timescales for geologically stabilized carbon. The IRF of Lashof and Ahuja (1990) was removed from the manuscript as suggested by the reviewer.

- *Line 265: I disagree with the idea that changing u will, in general, not lead to a change in B. I think there is quite a bit of evidence (forest self-thinning, soil carbon saturation) that B is highly sensitive to changes in u in real ecosystems. This comes up again a few paragraphs later. While it is mathematically convenient to separate these two things, I think in general it is not really possible to*

*change without the other (nor a priori to assert what sign that change to u or B will necessarily be).*

This is basically the difference between a linear and a nonlinear system, and this is why we make the distinction in our manuscript. In a linear system $u$ and $\mathbf{B}$ are independent by the same definition of linearity. However, in many systems they are not independent as the reviewer points out, and this is a strong indication for nonlinear behavior. However, our intention in this section is to show how CS and CBS behave in the linear case, so when someone makes this assumption is aware of the consequences. We added text in this section making this point clear. We try to show that these results only apply to linear systems in equilibrium, but for more realistic systems this is not the case. We also want to point out that even though the assumptions may be unrealistic, they are still made in many different analyses and for this reason it is important to know what are the consequences of the linearity assumption.

- *Line 293: This is a fairly obvious result and so I'm not sure why this formalism is needed to make that point?*

Yes, this is an obvious result, but there are policy relevant cases in which this is not so obvious. For example, in discussions of the 4 per mil initiative it is commonly assumed that inputs of carbon to soils can be achieved by increasing C inputs by a proportion of 0.004 of the current soil carbon stocks. However, there is no distinction between increasing C sequestration by management inputs versus management rates, or a combination of both. Therefore, we believe it is still important to clearly show that carbon sequestration can be maximized by managing both. We do not show here any formal optimization analysis, but the idea is that this framework can be used to better pose the maximization problem on formal mathematical grounds. We elaborate now better on this idea in the new version.
- *Section 5. I am not sure I understand the point of this example, and I also think there is a conceptual error being made here when the method is applied to large (relative to the total biospheric fluxes) sequestration perturbations: as I understand the notation used here, the function h_a(t) represents the remaining pulse (positive or negative) of CO2 into the atmosphere. But much (âĹij50%) of the loss of that atmospheric concentration pulse is due to the beta effect of land carbon responding to the carbon that has been emitted. So it seems like you are double counting this biospheric response, as it appears in both h_a(t) and in r(t)? I suspect this whole approach only works for small perturbations to the biosphere, where h_a(t) and r(t) are approximately non-overlapping, and thus excludes this example here.*

The idea behind this example was simply to show that the CBS and the CS metrics can be computed for a time-dependent situation. The reviewer is correct in that for a large perturbation, there is potential for double counting because the atmospheric response already includes biospheric effects. For a correct computation of the time-dependent response of the atmosphere to large biospheric perturbations, a time-dependent response function $h_a(t_0, t - t_0)$ obtained directly from the particular simulation should be used. This function should exclude the effects of the biosphere and only include carbon removal from ocean sinks. However, since our aim is simply to show how to compute CBS for the time-dependent case, we decided to remove this example. We use now a different model that works at the ecosystem level and not at the biosphere level. With this model we show now how to compute the proposed metrics for systems out of equilibrium within the limit of a small perturbation, so we can still use a constant atmospheric response function.

- *Table 1. Where are these numbers coming from? It seems like the authors are just sort of making them up as heuristic examples. Is that the case, and if so, might it be more useful to use numbers based on real-world, even if highly sim-*

*plified, examples? Similarly, the u and B numbers from Emanuel et al (1980) are for a globally-averaged ecosystem? If so, I think this wouldn't make sense for this example and you would have to use examples for a specific forest ecosystem instead. I understand the intention here is to be heuristic but more realistic numbers shouldn't be too hard to track down and it'd be infomrative to try to do something that corresponds more to reality when talking about concrete examples such as this.*

These are indeed heuristic values, only for showing the consequences of different types of biospheric carbon management. However, for the reasons mentioned earlier, we decided to remove this example.

- *Line 395. Can they? I see how albedo could, but other surface energy terms or surface roughness imply a tradeoff of one or another type of energy, or redistributions of energy between the land and atmosphere, and so can't really be compared to this metric.*

They cannot be compared directly, and we only claim that they are in 'units more comparable to those used to assess the overall effect of forest on climate'. The point is that values of CBS in units of W m$^{-2}$ yr may be easier to relate to energy balance terms than GWP values, which are reported in $CO_2$-equivalents.

- *Section 7: I'm not totally sure how comparing the CBS metric of two extremely vintage ecosystem models (one of which is a global-mean number and the other is a sort of reference-temperature number, so not really comparable even) is really of any importance to the argument being made in this manuscript, or anything else really. If the point is just that soils store a lot of carbon for a long time, don't we already know that? Suggest substantially revising or deleting this section.*

This section/example was removed as recommended by the reviewer.

- *Line 476-484. The other (much larger, really) problem with GWP is that its utility completely depends on what time interval the metric is integrated over; hence*

*the unending debates about how much policy should focus on CH4 as compared to CO2. This problem applies equally to the CBS, but is completely skipped over in this paper. How would the CBS be used in a policy-relevant context where we care about limiting peak temperature at some time period? Are there sequestration methods that are positive at a 50-year time horizon but negative on shorter or longer timescales? Is there a CBS analogue for GWP-star? Exploring these question would seem to be central to how this metric would actually be used in practice, but isn't actually touched on at all in this manuscript. I think this is a mistake and a major shortcoming of the current manuscript.*

Thanks for this suggestion. There has been a lot of debate on the time horizon for integration in GWPs, and this is indeed problematic for the case of emissions. But for the case of sequestration, it is an advantage to consider a finite time period for assessing sequestration. This is basically the problem of Permanence in the carbon accounting literature, where it is clear that sequestrations of carbon cannot be considered as permanent. To address this topic we decided to add an example with differences in integration time to show that different conclusions could be obtained by comparing systems at different integration times. We also added a section in the Discussion on this topic.

- *Lines 501-509. These are really not trivial problems, and substantially degrade the utility of this metric. The criticism of the Joos et al model strikes me as wildly off base; the irreversibility of global warming on shorter than multi-millenial timescales is a core feature of the problem and so asserting it away as something that can be ignored is not a good idea. In principle, the uncertainty in the impulse response function would be the same if used for two separate treatments, i.e. a baseline and a perturbed ecosystem, thus it seems like the more useful application of this method would be as an analog to GWP (not AGWP): calculate CBS of both a directly-perturbed ecosystem and an unperturbed ecosystem (or relatively unperturbed, i.e. not logged or afforested or whatever the treatment is, but*

*still subject to CO2 fertilization, changes in climate-driven mortality, etc), recognizing that, in a globally changed world, neither will likely be at steady state, and calculate a relative CBS as the ratio of the two absolute CBS.*

Again, we reconsidered this point and believe this manuscript is not the right venue to challenge the IRFs of Joos et al. (2013), so we removed this text from the manuscript. Instead, we take the same simple approach of replacing the infinity time scale in Joos' IRF and replaced it by a 1 million year timescale as in Millar et al. (2017). This removes the mathematical problem of finding a limit to our integrals and has no practical consequence for policy relevant timescales. We do appreciate the suggestion of the Reviewer about computing a relative metric to compare CBS for two cases, e.g. a perturbed and unperturbed system. We did something very similar in the forestry examples in the previous version of the manuscript, but instead of computing a ratio we computed a difference between the two CBS values. In the new version of the manuscript, we added now a computation of the ratio of CBS between two systems, and added a discussion about how one could use this ratio for problems similar as in the use of GWPs.

**References**

Joos, F., Roth, R., Fuglestvedt, J. S., Peters, G. P., Enting, I. G., von Bloh, W., Brovkin, V., Burke, E. J., Eby, M., Edwards, N. R., Friedrich, T., Frölicher, T. L., Halloran, P. R., Holden, P. B., Jones, C., Kleinen, T., Mackenzie, F. T., Matsumoto, K., Meinshausen, M., Plattner, G.-K., Reisinger, A., Segschneider, J., Shaffer, G., Steinacher, M., Strassmann, K., Tanaka, K., Timmermann, A., and Weaver, A. J. (2013). Carbon dioxide and climate impulse response functions for the computation of greenhouse gas metrics: a multi-model analysis. *Atmospheric Chemistry and Physics*, 13(5):2793–2825.

Lashof, D. A. and Ahuja, D. R. (1990). Relative contributions of greenhouse gas emissions to global warming. *Nature*, 344(6266):529–531.

Millar, R. J., Nicholls, Z. R., Friedlingstein, P., and Allen, M. R. (2017). A modified impulseresponse representation of the global near-surface air temperature and atmospheric concentration response to carbon dioxide emissions. *Atmospheric Chemistry and Physics*, 17(11):7213–7228.

---

## Referee Report (RR1)

My Review

This manuscript presents a new metric that the authors argue accounts for carbon sequestration better than established metrics such as absolute global warming potential. Considering the implications this metric could have on policy it is extremely relevant. However as presently written Sierra et al. is unclear and falls short of meeting its objectives. Overall this is an interesting paper that would greatly benefit from revisions.

The following should be addressed
1. Throughout the manuscript the authors suggest that CBS could be used in place or complimentary to GWP because GWP fails to take into account carbon sequestration and how it varies between ecosystems. On a "gut" level this makes sense, however the authors fail to provide concrete numerical evidence that CS and CBS vaires between ecosystems, and/or that these differences matter at the global scale.
2. It is unclear if the CS/CBS results presented in the manuscript are calculated on a global scale or as an aggregate of different ecosystems. For example in section two the authors present equation 29, "X is a vector of ecosystem carbon pools" but fail to discuss how many ecosystems are modeled, which ones, and where the parameterizations come from.

3. In section 3.2 the use of increase (decrease) and decrease (increase) relating to different carbon management policies is confusing starting in lines 302. Is this notation saying that the carbon storage is either increasing or decreasing? Or are they referring to the rate of change of the decreasing carbon inputs?

Other specific comments

L 45: are the authors suggesting that daily carbon sequestration can impact atmospheric CO2?

L 236: please provide some more information about TECO, not all readers will be that familiar with it, is it a global model or regional model? How many ecosystems does it represent?

L 300: Does management include the global anthropogenic increase in CO2 concentrations? Or is it only concerned with ecosystem inputs?

---

## Author Response (AR2)

**Max–Planck–Institut für Biogeochemie**

Max Planck Institute for Biogeochemistry

Max-Planck-Institut für Biogeochemie

MPI für Biogeochemie · Postfach 10 01 64 · 07745 Jena, Germany

Dr. Jens-Arne Subke Associate Editor Biogeosciences

Dr. Carlos A. Sierra Research group leader Tel.: +49-(0)3641-57-6133 csierra@bgc-jena.mpg.de

4th December 2020

Dear Jens,

Thanks for the opportunity to submit a revised version.

**Comments from editor**

I have received three reports for your manuscript. These are generally supportive, and I would like to encourage you to address the points raised by referees. One referee report was not uploaded to the editorial system, but I include the comments below with apologies for this slightly unconventional way of getting these to you.

Whilst the referees raise a number of issues, I think that these can be addressed in another round of revisions, and would like you to address the points by referee 1 and referee 3 (below).

Thanks for the opportunity to reply to these reviewers' comments. We made two main changes based on these comments, 1) we modified the model in the example to allow the inputs to enter the system in the form of gross primary production and not as net primary production. Reviewer 1 was right about this problem, which is a problem of the original model and not of our mathematical framework. To solve the issue without changing the model again, we simply added back the autotrophic respiration component to the input flux and removed it from the biomass of the autotrophic pools. More details are in the answers to Reviewer 1. 2) Following the suggestion from Reviewer 2, we added an example in the supplementary material in which we show how to compute CS and CBS for nonlinear models out of equilibrium.

Additional minor changes are described below in the answers to all comments.

**Comments from referee 1**

The manuscript submitted by Sierra et al introduces the maths behind a useful metric for quantifying The Climate Benefit of Carbon Sequestration as the title suggests. The work focuses on a simple concept that needs to be understood more broadly that a unit of carbon removed from the atmosphere by net ecosystem productivity in a given year does not have an infinite lifetime in that ecosystem and that the timing of its return to the atmosphere should be factored into the accounting of climate benefits. Indeed, this time element of accounting is critical

Max-Planck-Institut für Biogeochemie Hans-Knöll-Straße 10 07745 Jena Germany Tel.: +49-(0) 3641 / 57-6110 Fax.: +49-(0) 3641 / 57-6110 http:// www.bgc-jena.mpg.de Direktorium Susan Trumbore Markus Reichstein (Managing Dir.) Sönke Zaehle ID-Nr. DE 129517720

for a full assessment of the true climate benefits of emissions and sequestration, and has been particularly underappreciated when assessing climate change mitigation opportunities via a sequestration pathway.

This idea is not new to those with expertise in the coupled carbon-climate system, but the introduction of this quantitative metric provides a new tool in the toolbox that should improve the work of scientists, policymakers and managers alike. Unfortunately, the paper does a poor job of displaying and communicating the application of this metric for real-world situations. As previous reviewers have noted, the examples are hard to follow and are not clearly illustrative. It would be nice to see the manuscript revised to make the examples more accessible. This is not a requirement for the current paper to make a contribution, but some mostly subtle changes would lay the foundation for a broader audience to understand and perhaps adopt this new approach. Also, a few errors need to be corrected.

Thanks for recognizing the importance of the manuscript. We acknowledge that the examples are difficult to follow, but we also think they are improved compared to the previous version of the manuscript. To address this concern, we added additional text for each example better explaining the context and the implications for each case.

Major concerns are listed in order of presentation in the paper, not their importance, but \*\* has been placed in front of the most important concerns.

P6, L148: The timing of carbon release from ecosystems to the atmosphere often involves stochastic processes in the real world, such as with punctuated events of windthrow-driven tree mortality or from wildfires. The simple model of ecosystem carbon flows and ensuing ecosystem respiration is not capable of accommodating this stochasticity. While acceptable in the current form, mention of this here is warranted, as is some discussion of this in the discussion section.

The model we used in the examples does not capture stochastic processes such as those related to disturbances, but the overall mathematical framework we introduce in Section 2, and particularly in equation 8, can deal with these stochastic processes. The reason for this is that the state transition operator  $\Phi(t, t_0)$  captures the complete fate of carbon in the ecosystem, from fixation until final release, independently on whether this release is due to respiration or fires. In the sentence mentioned by the reviewer, we say: "(mostly ecosystem respiration)", but the release can happen due to other reasons. We modified this sentence to clarify this point.

P6, L163 / Eq 10: Is the IRF for emission reversible, that is, is it invertible for carbon removals as sequestration? This property is implied by the approach in Eq 10, yet the underlying biological and physicochemical processes might not behave in exactly this way. I suspect they are, particularly for ocean uptake / release, and agree that this is the standard assumption (one I too have relied upon). However, for the land response I am more suspicious and seeing it written here made me wonder. Land biospheric uptake of CO2 is not symmetric with release of CO2 so I suspect that at least that portion of the IRF is not exactly reversible.

As the reviewer points out, this is the standard assumption in dealing with removals from the atmosphere. It is a direct consequence of the assumption of

**Max–Planck–Institut für Biogeochemie**

To Dr. Jens-Arne Subke

4th December 2020

'sequestration as a negative emission'. We acknowledge that there is still uncertainty on whether this is a reasonable assumption for land sequestration, but we refrain from questioning this standard assumption because it would be out of the scope of the current manuscript.

P10, L30: As noted by R2, uptake rate (u) and transfer fates (B) are not independent in many real-world ecosystems (forests make an easy example). Some discussion of the degree to which this alters (or undermines?) the current simplified approach is warranted, with mention here and a 2 to 3 sentences in the discussion.

Again, it is important to distinguish here between the simple model used for the example versus the overall framework presented in section 2. We added additional text here mentioning that this is an assumption of the model, and not of the overall framework. We mention this topic again in the discussion.

\*\* Throughout, the papers use of the term uptake seems badly misleading. NPP is not uptake, and u is carbon inputs to ecosystems in the sense of net productivity. This is wholly different from ecosystem uptake and this oversight is surprising and even embarrassing given the subject of this manuscript, which is to point out that sequestration benefits need to separate carbon inputs from carbon releases. See more in the next comment.

We agree with the reviewer on this point, and have to acknowledge that the only reason why we use NPP as uptake was due simply to the choice of model for the example. Many models make the assumption that after photosynthesis, autotrophic respiration is immediately removed from the ecosystem, and then NPP is allocated to vegetation pools. We completely agree with the reviewer in that this is incorrect and misleading in the context of this manuscript, but it is hard to find a simple model in the literature that does not make this assumption so we can use in our example. To address this issue, we modified the model structure of the TECO model by defining uptake u as GPP, and removing autotrophic respiration from the vegetation pools after allocation, and not before as in the original model. This change to the model structure, changes all numerical results we present in the revised version of the manuscript, but the shapes of all curves and all general trends remain unaltered.

\*\* P10, L261: The nature of the experiment in Fig 4 needs to be clarified. Is this a pulse experiment or a step experiment? The caption describes the experiment as instantaneous carbon uptake at any given time, but what that means is puzzling. Perhaps I have misunderstood. It sounds like a pulse experiment but that does not seem correct. Instead, the experiment appears to involve a sustained boost in uptake rate (NPP?) that leads to a steady-state carbon balance as ecosystem stocks accumulate in response to the sustained elevation of NPP and then ultimately lead to a balance of inputs and outputs. However, NPP is not ecosystem uptake, and it is badly misleading to define it as such. In fact, I suggest modifying the symbology and the name of this term to be p, productivity, or i, inputs. Expand the figure to show a time series of u, your so-called uptake, a time series of r respiration, a time series of their net balance, and also the pools over time. Including such figures for each of the experiments will greatly improve clarity and improve the communication of the time-dynamics of the CBS metric.

**To Dr. Jens-Arne Subke**

The results presented in Fig 4 were not obtained from a pulse experiment, a step experiment, or any form of forward simulation run. They were obtained using the formulas presented in section 2.4 after computing the steady-state solution analytically, and not from a simulation. In particular, we used the state transition matrix defined as  $\Phi(t, t_0) = e^{a\mathbf{B}}$ , for which we already know **B** from the model equations, so we can compute the matrix exponential directly without the need to perform a pulse-response experiment. We can also compute the steady-state directly as  $x^* = -\mathbf{B}^{-1}u$ . In other words, we do not need to run simulations as it is done in traditional modeling studies. Therefore, we cannot present time-series of inputs and outputs because this is not the approach we followed. However, we acknowledge that the presentation of these results can be improved. We expanded the paragraph to explicitly mention the approach and the equations used to obtain the results.

\*\* Fig 4 and 6: The captions suggests that the experiment involves a fixed total uptake of 6.64 MgC ha-1 that lead to a steady-state carbon stock response of +240ish Mg C. That is illogical. Is the uptake meant to be an annual input of carbon to the ecosystem? Must be. The units should be 6.64 MgC ha-1 yr-1.

Please note that we are following a sequestration pulse. In this case, we use the amount of inputs that enter the ecosystem in one year, but the unit is still in MgC  $ha^{-1}$  because we are not following a continuous input flux, only the 'instantaneous' amount that enters in one year. We modified the captions to make this point clear.

\*\* P10, L271: It looks to me as though the labels might be wrong for the PI100 and PD100 curves in Fig 4c. Removing CO2 from an atmosphere that has more CO2 should cause a smaller cooling effect than the same removal applied to an atmosphere that has less CO2. Havent I got this right?: The radiative efficiency of CO2 decreases with increasing CO2 concentration. Your results seem to suggest the revers in Fig 4, though Fig 5 looks correct.

Thanks for pointing this out. There is indeed a problem with the labels, but not in Fig 4c, but in Fig 5, opposite to the comment by the reviewer. The reason for the difference can be well explained by this excerpt from Joos et al (page 2809, section 4.4.1): The lower  $CO_2$  perturbation for PI100 is generally due to a higher uptake by both the ocean and the land biosphere and is consistently lower for PI than PD conditions for all individual models. What happens here is that ocean and land uptake override the opposite radiative efficiency effect in all models used by Joos et al. In consequence, the removal applied in present day conditions shows a larger CBS (more cooling) than in preindustrial conditions because during PI carbon removals are proportionally higher and additional sequestration has less impact than during PD.

We fixed the problem with the labels in Fig 5. and discuss the issue in more detail on page 10.

\*\* P15, L417: R2 correctly pointed out that it is badly misleading to suggest that CBS W m-2 could be directly compared to the terms in the surface energy balance that also have W m-2. This sentence must be removed or rephrased. A top-of-atmosphere radiative forcing is comparable to the CBS measure but not a surface energy flux perturbation. For example, a change in latent energy flux of X W m-2 does not have a comparable effect on the planetary energy budget, only on the local surface, and the associated energy is returned to the lower atmosphere as heat

**Max–Planck–Institut für Biogeochemie**

To Dr. Jens-Arne Subke

when that water vapor is condensed. This statement contributes to a continuing misunderstanding even among scientists that really needs to be corrected.

This statement was removed as suggested.

Minor Comments:

P3, L 69 / Eq 1:  $K_{co2}$  is also time varying as it depends on atmospheric composition changing over time and as it influences the radiative efficiency of any particular GhG. This is noted on P4, L100 but it struck me as missing already here.

We introduce the time dependency in equation (1) as suggested, with a reference to section 2.2 for details on the time-independent assumption.

Figures 4 and 5: In the figure caption, write out the definitions of PI100 and PD100.

Definitions added to captions as suggested.

To Dr. Jens-Arne Subke

4th December 2020

**Comments from referee 2**

The revised manuscript by Sierra et al has a cleaner focus on the main points of their argument, and is thus I think a better and clearer manuscript. My concerns from the prior version are mostly satisfied, and I think the paper makes an important contribution. However, one of my concerns still remains: does the approach work for the autonomous, nonlinear, nonequilibrium case? All of the examples are for the linear autonomous case (both equilibrium and non-equilibrium), and the authors state that it mathematically can't work for the nonautonomous cases, but what about the autonomous nonlinear cases, equilibrium and non-equilibrium: does the method work for these or does it not? And if the method does apply in the nonlinear case, please provide an example. This is highly relevant to real world problems, so it would be good to state and show clearly in the manuscript what the bounds of the technique presented here are.

The main equations developed in section 2 can be applied to linear and nonlinear models, autonomous and non-autonomous, as long as the state-transition matrix  $\Phi(t,t_0)$  can be computed. In the manuscript, we work only with a linear autonomous model for which the state transition matrix is equal to the exponential of the compartmental matrix. For nonlinear systems the procedure is more complex, but it is possible to obtain the state transition matrix based on methods we presented in Metzler et al. (2018, PNAS 115: 1150). To address this concern from the reviewer, we added an example in the supplementary material where we take a two-pool soil microbial model with a strong nonlinear interaction. The model is the same as the one presented in Wang et al. (2014, Biogeosciences 11: 1817), which shows very strong oscillations due to the nonlinearities. We computed CS and CBS for this model using a Python code we developed. This supplementary material is presented in the form of a Jupyter Notebook, so interested readers can reproduce our computations and adapt them to other nonlinear models.

To Dr. Jens-Arne Subke

4th December 2020

**Comments from referee 3**

This manuscript presents a new metric that the authors argue accounts for carbon sequestration better than established metrics such as absolute global warming potential. Considering the implications this metric could have on policy it is extremely relevant. However as presently written Sierra et al. is unclear and falls short of meeting its objectives. Overall this is an interesting paper that would greatly benefit from revisions. The following should be addressed:

1. Throughout the manuscript the authors suggest that CBS could be used in place or complimentary to GWP because GWP fails to take into account carbon sequestration and how it varies between ecosystems. On a gut level this makes sense, however the authors fail to provide concrete numerical evidence that CS and CBS vaires between ecosystems, and/or that these differences matter at the global scale.

We based our argument about the difference between GWP and CBS on the formal mathematical definition of these metrics. Equation (1) clearly shows that AGWP quantifies warming for carbon that enters the atmosphere. In equation (11) we define CBS as the atmospheric effect for carbon that enters an ecosystem and returns to the atmosphere. In Figures 4 and 5 we show numerical differences between these metrics. These are formal grounds for our arguments and not 'gut level' suggestions. Furthermore, our examples in section 3.2 (Figures 7 and 8) show differences of CBS for different cases that could be interpreted as different ecosystems with differences in photosynthetic and process rates.

We added a paragraph at the end of section 3.2 to make the point that these different cases in which we changed the vector of inputs u and the matrix of process rates **B** can be also interpreted as different ecosystems. In addition, we added references to other studies that quantify mean transit time in ecosystems to highlight that transit time can vary substantially among ecosystems and therefore CS and CBS are expected to change considerably in the terrestrial biosphere with implications at the global scale.

2. It is unclear if the CS/CBS results presented in the manuscript are calculated on a global scale or as an aggregate of different ecosystems. For example in section two the authors present equation 29, X is a vector of ecosystem carbon pools but fail to discuss how many ecosystems are modeled, which ones, and where the parameterizations come from.

The examples only apply to a particular ecosystem representative of the Duke Forest in North Carolina, USA. The model we used was parameterized for this ecosystem. This is explained in the first paragraph of section 3.1 and parameter values are giving in the Appendix.

We added extra references later in the examples to remind the reader that the examples are for a temperate forest representative of the Duke Forest.

3. In section 3.2 the use of increase (decrease) and decrease (increase) relating to different carbon management policies is confusing starting in lines 302. Is this notation saying that the carbon storage is either increasing or decreasing? Or are they referring to the rate of change of the decreasing carbon inputs? We removed the notation 'increase (decrease)' using parenthesis and replaced it by additional sentences that explicitly explain the effect of increasing or decreasing inputs and process rates.

Other specific comments

• L 45: are the authors suggesting that daily carbon sequestration can impact atmospheric CO2?

Yes, carbon sequestration removes carbon from the atmosphere and during the time it is stored in an ecosystem it is removed from the radiative effects that produce warming. Without terrestrial carbon sequestration, a lot more  $CO_2$  would accumulate in the atmosphere than what currently accumulates. Maybe we do not understand well this comment, but we think this aspect of the carbon cycle is well understood.

- L 236: please provide some more information about TECO, not all readers will be that familiar with it, is it a global model or regional model? How many ecosystems does it represent? We added more details in the main text. Appendix C has a more detailed description of the model.
- L 300: Does management include the global anthropogenic increase in CO2 concentrations? Or is it only concerned with ecosystem inputs? In the computations we present, CS and CBS are quantified accounting for the return of CO2 to the atmosphere. If management results in more CO2 emissions, CS and CBS would capture this effect. Because the example in section 3 assumes equilibrium conditions, the effect of increased anthropogenic CO2 in the atmosphere is not considered. However, in sections 4.1 and 4.2, the model does consider the effects of increased CO2 concentrations in the atmosphere by increasing GPP proportionally, simulating a fertilization effect.

We hope that our responses and the new version of the manuscript satisfactory address previous concerns.

Thanks,

Carlos A. Sierra, PhD On behalf of all authors

**The Climate Benefit of Carbon Sequestration**

Carlos A. Sierra1, Susan E. Crow2, Martin Heimann1,3, Holger Metzler1, and Ernst-Detlef Schulze1

1Max Planck Institute for Biogeochemistry, 07745 Jena, Germany

2University of Hawaii Manoa, Honolulu, HI 96822, USA

[revised manuscript text omitted]

---

## Author Response (AR3)

**Max–Planck–Institut für Biogeochemie**
Max Planck Institute for Biogeochemistry

[Figure]

MPI für Biogeochemie · Postfach 10  01  64 · 07745 Jena, Germany

Dr. Jens-Arne Subke
Associate Editor
Biogeosciences

**Dr. Carlos A. Sierra**
**Research group leader**
Tel.: +49-(0)3641-57-6133
csierra@bgc-jena.mpg.de

29th December 2020

Dear Jens,

We are submitting here a new revised version of our manuscript with changes based on your suggestions. In particular, we fixed typos and clarified references to the Appendices, and added extra information regarding the supplementary material. You can see the new changes in the attached file with track-changes markup.

We hope the manuscript is now ready for publication. Thanks for your editorial work,

Best regards,

Carlos A. Sierra, PhD
On behalf of all authors

Max-Planck-Institut für
Biogeochemie
Hans-Knöll-Straße 10
07745 Jena
Germany

Tel.: +49-(0) 3641 / 57−6110
Fax.: +49-(0) 3641 / 57−6110
http:// www.bgc-jena.mpg.de

Direktorium
Susan Trumbore
Markus Reichstein (Managing Dir.)
Sönke Zaehle
ID-Nr. DE 129517720

[revised manuscript text omitted]